# Osteocalcin expressing cells from tendon sheaths in mice contribute to tendon repair by activating Hedgehog signaling

Yi Wang[1], Xu Zhang[1], Huihui Huang[1], Yin Xia[1], YiFei Yao[2], Arthur Fuk-Tat Mak[2], Patrick Shu-Hang Yung[3], Kai-Ming Chan[3], Li Wang[4], Chenglin Zhang[4], Yu Huang[4], Kingston King-Lun Mak[1]*

[1]Developmental and Regenerative Biology, School of Biomedical Sciences, The Chinese University of Hong Kong, Shatin, Hong Kong; [2]Division of Biomedical Engineering, The Chinese University of Hong Kong, Shatin, Hong Kong; [3]Department of Orthopaedics and Traumatology, The Chinese University of Hong Kong, Prince of Wales Hospital, Sha Tin, Hong Kong; [4]Neural, Vascular and Metabolic Biology, School of Biomedical Sciences, The Chinese University of Hong Kong, Shatin, Hong Kong

**Abstract** Both extrinsic and intrinsic tissues contribute to tendon repair, but the origin and molecular functions of extrinsic tissues in tendon repair are not fully understood. Here we show that tendon sheath cells harbor stem/progenitor cell properties and contribute to tendon repair by activating Hedgehog signaling. We found that *Osteocalcin* (*Bglap*) can be used as an adult tendon-sheath-specific marker in mice. Lineage tracing experiments show that *Bglap*-expressing cells in adult sheath tissues possess clonogenic and multipotent properties comparable to those of stem/progenitor cells isolated from tendon fibers. Transplantation of sheath tissues improves tendon repair. Mechanistically, Hh signaling in sheath tissues is necessary and sufficient to promote the proliferation of *Mkx*-expressing cells in sheath tissues, and its action is mediated through TGFβ/Smad3 signaling. Furthermore, co-localization of GLI1[+] and MKX[+] cells is also found in human tendinopathy specimens. Our work reveals the molecular function of Hh signaling in extrinsic sheath tissues for tendon repair.

DOI: https://doi.org/10.7554/eLife.30474.001

*For correspondence:
kingstonmak@gmail.com

Competing interests: The authors declare that no competing interests exist.

## Introduction

Tendon injuries, commonly caused by overuse or age-related degeneration, are hard to repair and represent a major clinical challenge. Mechanisms responsible for the reparative process remain elusive. It has been hypothesized that tendon heals through both intrinsic and extrinsic mechanisms in which cells from both within the tendons and the surrounding tissues are necessary for the healing process (*Chang et al., 1997*; *de Mos et al., 2007*; *Gelberman et al., 1984*; *Harrison et al., 2003*; *Jones et al., 2003*; *Manske and Lesker, 1984*; *Peacock, 1965*; *Potenza, 1962*; *Salingcarnboriboon et al., 2003*). However, molecular evidence for the contribution of extrinsic tissues to support such notion is missing.

Tendon fibers are enveloped by a dense membrane of connective tissues that are composed of epitenon and tendon sheaths, bridging blood vessels, lymphatics and nerves to tendon fibers (*Elliott, 1965*; *Hess et al., 1989*). Tendon sheaths are classified as extrinsic tissues in contrast to tendon fibers and epitenons, which are intrinsic structures (*Kannus, 2000*). Tendon sheath only covers tendon fibers in high friction areas where it provides lubrication for the functions of fibers comprising, for examples, the Tibialis anterior tendon, the peroneus longus and brevis tendon, and the

flexor and extensor tendons. *Tubulin polymerization-promoting protein family member 3* (*Tppp3*) is the first and only molecular marker to be identified that is expressed in developing epitenon and tendon sheath (*Staverosky et al., 2009*). However, *Tppp3* expression is also found in developing muscles and forming joints. It is not clear whether *Tppp3* is also expressed in adult tendon sheaths. Additional molecular markers are therefore needed to investigate the function of sheath tissues in adult tendon remodeling and repair.

Hedgehog (Hh) signaling plays important roles in the skeletal system, in both bone and cartilage development and homeostasis (*Karp et al., 2000*; *Vortkamp et al., 1996*; *Mak et al., 2006, 2008a, 2008b*). Apart from bones and cartilages, it has been recently demonstrated that Hh signaling is also involved in tendon development (*Liu et al., 2012, 2013*; *Schwartz et al., 2015*). Hh-responding cells were found in the tendon insertion sites but not in the mid-substances during the perinatal stage (*Liu et al., 2012*; *Schwartz et al., 2015*), suggesting that Hh signaling plays a role in the process that transitions fibrocartilage to tendon fibers. In addition, the removal of Hh signaling in *Scleraxis* (*Scx*)-expressing cells reduces the biomechanical strength of the adult mouse tendons (*Liu et al., 2013*). This suggests that Hh signaling regulates early events in tendon fiber formation and remodeling. These accumulating evidence sources prompted us to hypothesize that Hh signaling is implicated in tendon repair. Here, we identified a new adult tendon sheath specific marker *Osteocalcin*, which is encoded by the *Bglap* gene. We established that *Bglap*-expressing cells from sheath tissues possess stem/progenitor cell properties and that they participate in tendon repair. More importantly, Hh signaling mediates its effect through TGFβ signaling to regulate the expression of *Mkx* and collagen I. Our studies demonstrate the first molecular evidence for the contribution of extrinsic sheath tissues during tendon healing and that manipulation of Hh signaling may be a therapeutic target for tendon repair.

## Results

### Osteocalcin as a tendon sheath marker

The molecular evidence of extrinsic tissues in tendon repair is unclear. This is partly due to the lack of specific markers for molecular tracing of these cell populations during the repair process. To access the biology of tendon sheath tissues during tendon repair, we first determined markers that are specifically expressed in mouse adult tendon sheaths. *Tppp3* is the only marker reported that is expressed in developing sheath tissues, epitenon and paratenon (*Staverosky et al., 2009*). Consistent with this work in mouse embryos, we found that *Tppp3* was also specifically expressed in adult tendon sheaths (*Figure 1A*). Interestingly, *Osteocalcin*, which is encoded by the *Bglap* gene also showed a specific expression pattern similar to that of *Tppp3* in the sheath tissues. QRT-PCR analysis revealed that *Tppp3* and *Bglap* were highly expressed in sheath tissues as compared to tendon fibers (*Figure 1B*). Thus, we asked whether *Bglap* can be used as an adult tendon sheath specific marker. We crossed *BGLAP-Cre* transgenic mice, a *Cre* line driven by the human BGLAP promoter (*Zhang et al., 2002*; *Mak et al., 2008a*), with a $Rosa26^{mT/mG}$ mouse line (*Muzumdar et al., 2007*) for lineage tracing of *Bglap*-expressing cells in tendon tissues. As expected, strong and specific GFP signals were detected in tissues surrounding the tendon fibers of the *BGLAP-Cre; $Rosa26^{mT/mG}$* mice (*Figure 1C*). Cross-sections of tendon tissues displayed highly specific GFP signals in the sheath tissues but not in tendon fibers (*Figure 1D*). To further verify that the GFP⁺ cells are highly specific to tendon sheaths but not tendon fibers, the sheath and fiber tissues of the *BGLAP-Cre;$Rosa26^{mT/mG}$* mice were dissected separately and subjected to cell-sorting for green fluorescence. We found that about 80% of the cells isolated from the sheath tissues were GFP⁺, of which approximately 1.5% had clonogenic capacity as shown by a colony-formation assay (*Figure 1— figure supplement 1*). Consistently, both *Tppp3* and *Bglap* were strongly expressed in the sorted GFP⁺ cells of the sheath tissues but poorly expressed in GFP⁻ cells in the tendon fiber tissues (*Figure 1E*). Thus, our data indicate that *Bglap* is specifically expressed in adult tendon sheaths and can be used as a tendon sheath specific marker.

### Tendon sheath-derived cells are clonogenic and multi-potent

To determine whether tendon-sheath-derived cells possess stem/progenitor properties, we performed a single-colony assay using cells isolated from the tendon sheaths (GFP⁺) and tendon fibers

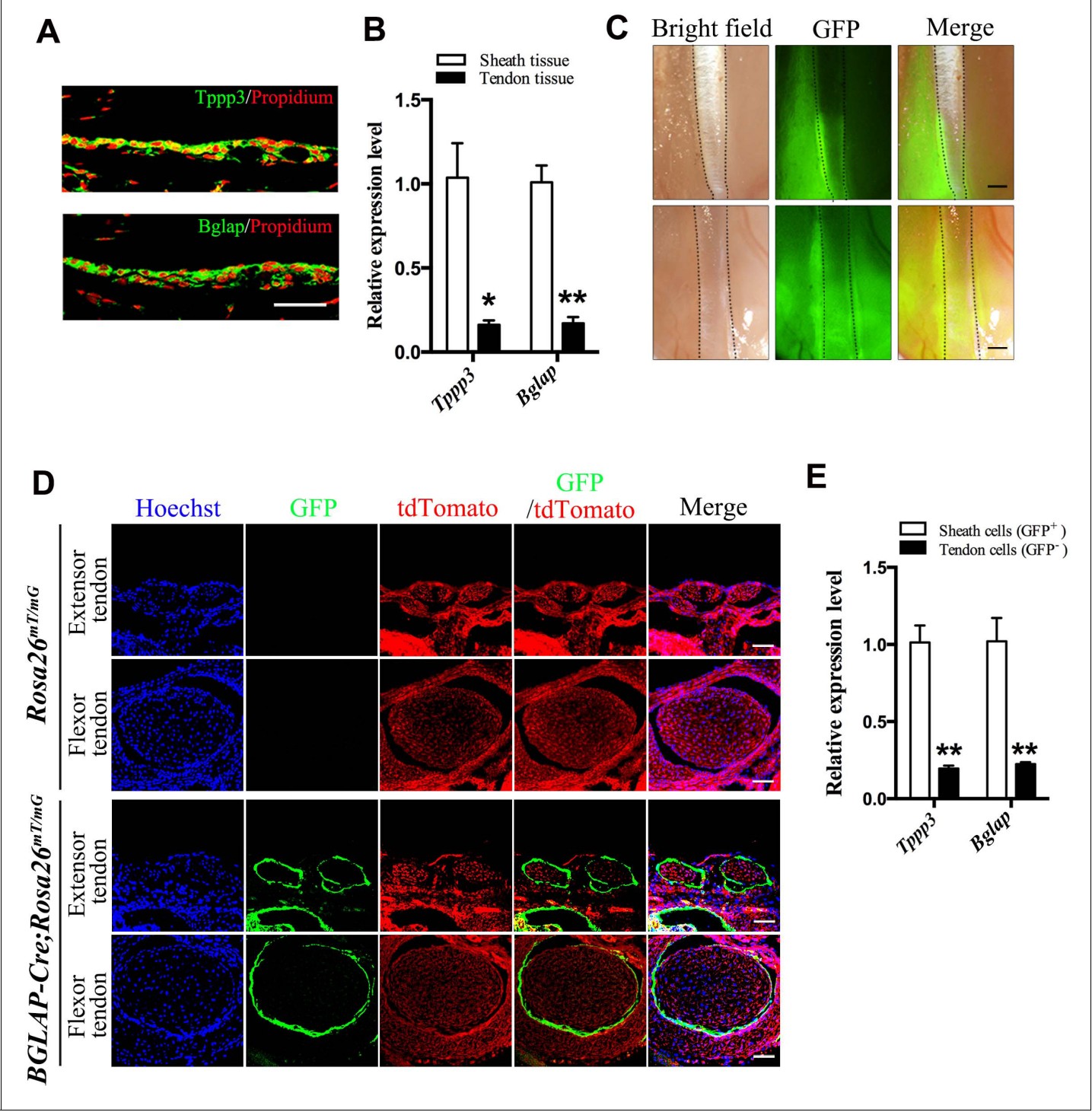

**Figure 1.** Bglap as a tendon sheath specific marker. (**A**) Fluorescent immunohistochemistry of mouse sheath tissues of the adult Tibialis anterior tendon at 2 months old (n = 5 mice per group). Scale bar, 40 μm. (**B**) QRT-PCR analysis of sheath-specific markers using adult mouse sheath and tendon tissues as shown. Relative expression levels were normalized to *β-tubulin* and the sheath tissues. Data are means ± s.e.m and were analyzed using Student's t-tests. *p≤0.013; **p≤0.0014. n = 3 mice per group. (**C**) Images of peroneus longus and brevis tendon (upper panel) and Tibialis anterior tendon (lower panel) of the *BGLAP-Cre;Rosa26^{mT/mG}* mice at 2 months old. Dotted lines depict the boundaries of the tendon fibers. GFP signals are observed in the tendon sheath tissues (n = 5 mice per group). Scale bar, 250 μm. (**D**) Cross-sections of fluorescent microscopic analysis of the extensor tendons and flexor tendons of *Rosa26^{mT/mG}* and *BGLAP-Cre;Rosa26^{mT/mG}* mice. Scale bar, 50 μm (n = 5). (**E**) QRT-PCR analysis of sheath-specific markers using sheath and tendon cells isolated from the *BGLAP-Cre;Rosa26^{mT/mG}* mice after cell-sorting for green fluorescence (GFP). Relative expression levels were

*Figure 1 continued on next page*

*Figure 1 continued*

normalized to *Gapdh* and the sheath cells. Data are means ± s.e.m and were analyzed using a Student's t-tests. **p≤0.0062. n = 3 biological replicates per group. Additional data for this figure are provided in *Figure 1—figure supplement 1*.

DOI: https://doi.org/10.7554/eLife.30474.002

The following source data and figure supplements are available for figure 1:

**Source data 1.** Source data relating to *Figure 1B*.
DOI: https://doi.org/10.7554/eLife.30474.005

**Source data 2.** Source data relating to *Figure 1E*.
DOI: https://doi.org/10.7554/eLife.30474.006

**Figure supplement 1.** Fluorescence-assisted cell-sorting (FACS) analysis of sheath-derived cells.
DOI: https://doi.org/10.7554/eLife.30474.003

**Figure supplement 1—source data 1.** Source data relating to *Figure 1—figure supplement 1B*.
DOI: https://doi.org/10.7554/eLife.30474.004

(tdTomato$^+$) of the *BGLAP-Cre;Rosa26$^{mT/mG}$* mice after cell-sorting. After 14 days, colonies derived from a single cell were visualized by fluorescence imaging and crystal violet staining (*Figure 2A,B*). Like tdTomato$^+$ tendon fiber-derived cells that possess stem-cell-like properties (*Bi et al., 2007*), GFP$^+$ sheath-derived cells also showed colonies of heterogeneous size and cell density. BMP2 and TGF-β1 are important factors in regulating bone and tendon formation (*Rodeo et al., 1999*; *Liu et al., 2015*; *Gori et al., 1999*; *Pryce et al., 2009*). Thus, we examined how sheath-derived cells responded to these treatments (*Figure 2—figure supplement 1A–C*). BMP2 and TGF-β1 induced the expression of tendon progenitor markers *Mkx* (*Ito et al., 2010*) and *Scx* (*Schweitzer et al., 2001*) (*Figure 2—figure supplement 1A*). However, only BMP2 but not TGF-β1 induced the osteoblast markers *Runx2* (*Komori et al., 1997*) and *Sp7* (*Nakashima et al., 2002*) (*Figure 2—figure supplement 1B*). Both BMP2 and TGF-β1 induced the expression of chondrocyte markers *Col2a1* (*Cheah et al., 1995*) and *Sox9* expression (*Lefebvre et al., 1997*) (*Figure 2—figure supplement 1C*). These responses were highly similar to those of tendon fiber-derived stem/progenitor cells (*Bi et al., 2007*).

Next, we tested the multipotent differentiation potentials of the sheath-derived cells toward osteogenesis, adipogenesis and chondrogenesis as compared to tendon-fiber-derived cells (*Figure 2C–H*). We found that sheath-derived cells differentiated into all three lineages with differentiating potential comparable to that of tendon stem/progenitor cells. All the lineage-specific markers were significantly increased under lineage-specific differentiation conditions (*Figure 2D,F,H*). To further determine the multipotent differentiation potential of sheath-derived cells *in vivo*, we subcutaneously transplanted the GFP$^+$ sheath cell sheets into the dorsal surface of the immunocompromised mice. After 8 weeks, the transplanted GFP$^+$ cell sheets formed tendon-like structures (*Figure 2I–L*). In some regions of the transplanted tissues, cells were stained positively for Alcian blue or Alizarin Red S (*Figure 2M,N*), suggesting that these cells also differentiated into chondrocytes and mineralized bone tissues. Similar results were observed using GFP$^+$ sheath cells mixed with matrix gels and subcutaneously injected into the calvaria of immunocompromised mice (*Figure 2—figure supplement 1D–I*). Collectively, our data reveal that Bglap$^+$ sheath cells harbor clonogenicity and multipotency similar to that of the tendon-fiber-derived stem/progenitor cells both *in vitro* and *in vivo*.

## Tendon-sheath-derived cells contribute to tendon repair

The mesenchymal properties of putative sheath stem/progenitor cells suggest that these cells contribute to tendon repair and regeneration. To investigate the repair capacity of tendon sheath *in vivo*, a full-thickness medial edge of the defect was created along the Tibialis anterior tendon of the *BGLAP-Cre;Rosa26$^{mT/mG}$* mice where true tendon sheath tissues surrounded the tendon fibers. At Day 14, GFP$^+$ cells appeared in the areas of injured tendon fibers as shown in both cross- and longitudinal-sections (*Figure 3A*, *Figure 3—figure supplement 1A*). Furthermore, these GFP$^+$ cells also co-expressed the tendon progenitor marker Mkx (*Figure 3B*, *Figure 3—figure supplement 1B*). These results suggest that sheath tissues possess stem/progenitor cells that differentiate into tenocytes for tendon repair. To further demonstrate the repair ability of sheath-derived cells, we isolated

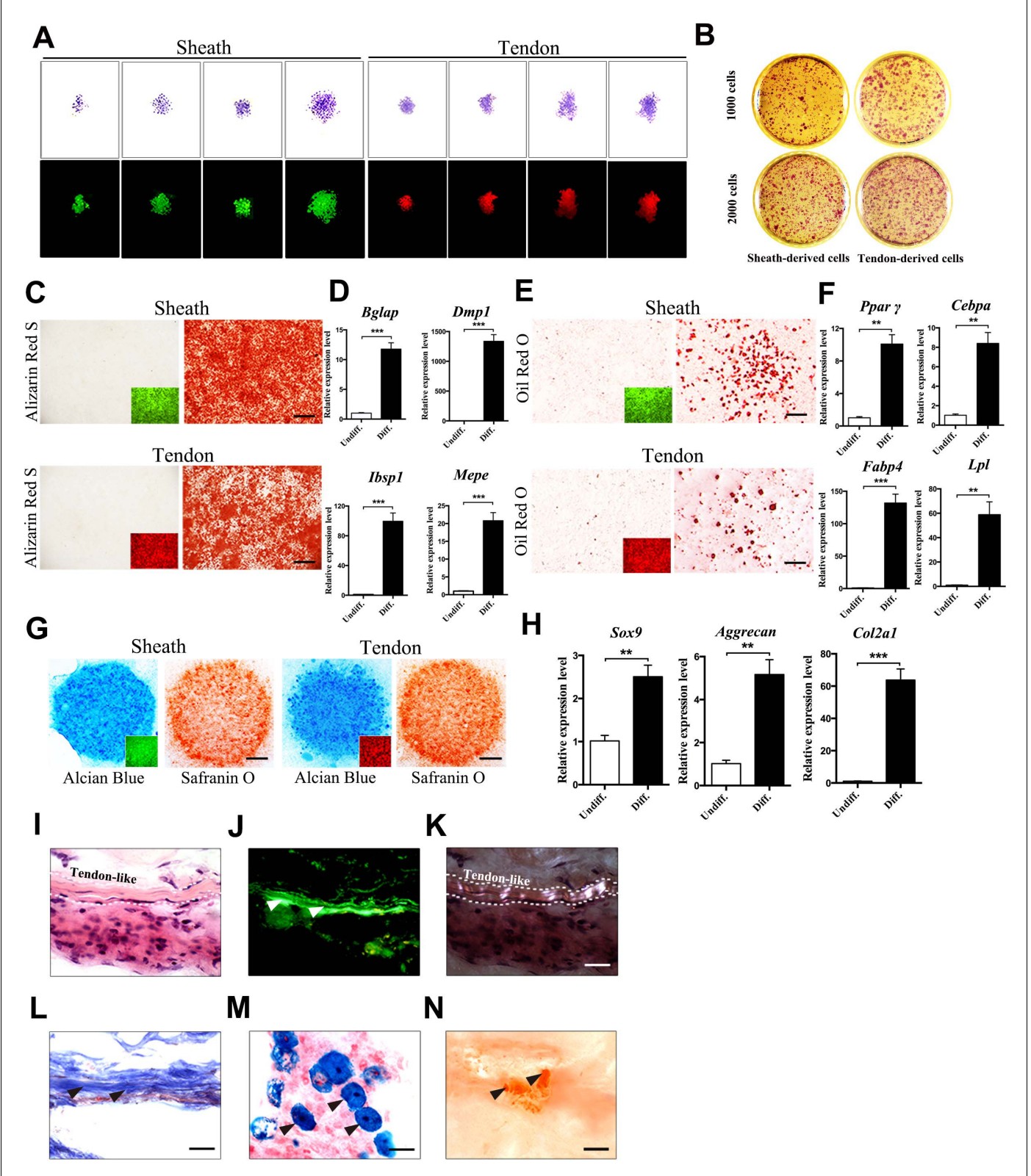

**Figure 2.** Multipotent differentiation potential of sheath-derived cells. (**A**) Colony formation assay using sheath-derived cells (GFP[+]) or tendon-derived cells (tdTomato[+]) isolated from the adult tendon tissues of *BGLAP-Cre;Rosa26[mT/mG]* mice. Colonies were stained with crystal violet (upper panel) or by fluorescence (lower panel) after 14 days of culture (n = 5). (**B**) Primary cells were seeded with two different initial cell densities of 1000 or 2000 for 14 days as shown. (**C, E, G**) Lineage differentiation assay under (**C**) osteogenic (scale bar, 1000 µm), (**E**) adipogenic (scale bar, 200 µm), and (**G**)

*Figure 2 continued on next page*

*Figure 2 continued*

chondrogenic conditions (scale bar, 1000 μm) using primary cells isolated from the tendon tissues of *BGLAP-Cre;Rosa26^{mT/mG}* mice after cell-sorting. Insets represent the fluorescence signals of the respective primary cells before differentiation. GFP^+ cells represent cells from sheath tissues; tdTomato^+ cells represent cells from tendon fibers (n = 3). (D, F, H) QRT-PCR analysis of gene markers for (D) osteogenesis, (F) adipogenesis and (H) chondrogenesis. Relative expression levels were normalized to *Gapdh* and the undifferentiated condition. Data are means ± s.e.m and were analyzed using Student's t-tests. **p≤0.0074; ***p≤0.001. n = 3 biological replicates per group. (I–N) Histologic analysis of tendon-like structures using GFP^+ sheath cell sheets transplanted into the dorsal surface of immunocompromised mice after 8 weeks. Tendon-like tissues were identified under (I) H&E staining, (J) Green fluorescence, (K) polarized light, and (L) Masson's trichrome staining. Arrowheads point to regions of (J) tendon-like structure, (M) chondrocytes as shown by Alcian blue staining or (N) mineralized bone as shown by Alizarin Red S staining. Dashed lines indicate the boundaries of tendon-like tissues (n = 5 mice). Scale bar, 20 μm. Additional data for this figure are provided in *Figure 2—figure supplement 1*.
DOI: https://doi.org/10.7554/eLife.30474.007

The following source data and figure supplements are available for figure 2:

**Source data 1.** Source data relating to *Figure 2D*.
DOI: https://doi.org/10.7554/eLife.30474.012
**Source data 2.** Source data relating to *Figure 2F*.
DOI: https://doi.org/10.7554/eLife.30474.013
**Source data 3.** Source data relating to *Figure 2H*.
DOI: https://doi.org/10.7554/eLife.30474.014
**Figure supplement 1.** Multipotent differentiation potential of sheath-derived cells *in vitro* and *in vivo*.
DOI: https://doi.org/10.7554/eLife.30474.008
**Figure supplement 1—source data 1.** Source data relating to *Figure 2—figure supplement 1A*.
DOI: https://doi.org/10.7554/eLife.30474.009
**Figure supplement 1—source data 2.** Source data relating to *Figure 2—figure supplement 1B*.
DOI: https://doi.org/10.7554/eLife.30474.010
**Figure supplement 1—source data 3.** Source data relating to *Figure 2—figure supplement 1C*.
DOI: https://doi.org/10.7554/eLife.30474.011

GFP^+ tendon sheaths from the *BGLAP-Cre;Rosa26^{mT/mG}* mice and transplanted them into the injured Achilles tendon of the left hind leg of immunocompromised mice in which the tendon is not surrounded by true tendon sheaths (*Kannus, 2000*). A full-thickness medial edge of the defect was created along the entire Achilles tendon and the sheath tissues transplants were placed in close proximity underneath the injured tendon. Two days after injury, most of the GFP^+ sheath tissues were still localized at the implanted position (*Figure 3C*). At Day 14, GFP^+ sheath tissues migrated towards the injured tendon fibers. By Day 45, GFP^+ cells were markedly found within the injured tendon fibers and had differentiated into tendon-fiber-like structures. Close examination revealed that the GFP^+ cells were morphologically, molecularly and biomechanically similar to tendon-like fibers (*Figure 3D–I*). The GFP^+ cells were polarized (*Figure 3D*) and thetendon progenitor markers *Mkx* and *Scx* (*Figure 3E*) as well as tendon extracellular matrix (ECM) components (*Figure 3F*) were strongly unregulated in the sheath-transplanted tendons. Furthermore, the overall collagen content was significantly increased in the sheath-transplanted group after injury (*Figure 3G*). Biomechanical testing of peak force, peak stress and stiffness was also significantly improved in the sheath-transplanted group (*Figure 3H and I*). Co-localization of Mkx expression with GFP^+ cells was also observed at the injured sites (*Figure 3J*, *Figure 3—figure supplement 2A*). In addition, FACS-sorted GFP^+ cells that were isolated from the grafts 14 days after the injury also displayed a molecular tendon signature (*Figure 3K*). Altogether, our results indicate that *Bglap*-expressing sheath cells contribute to tendon repair.

## Hh signaling promotes the proliferation of sheath progenitor cells

To dissect the mechanistic regulation that activates sheath stem/progenitor cells for tendon repair, we searched for signaling pathways that are implicated during early tendon cell differentiation. It has been previously reported that Hh signaling is activated for tissue regeneration (*Wang et al., 2015*; *Zhao et al., 2015*). In addition, Hh signaling is also active at the insertion site and in tendon cells during tendon development (*Liu et al., 2012, 2013*). Its activity correlates with cell proliferation rate in tendon tissues. These data suggest that Hh signaling may control the early

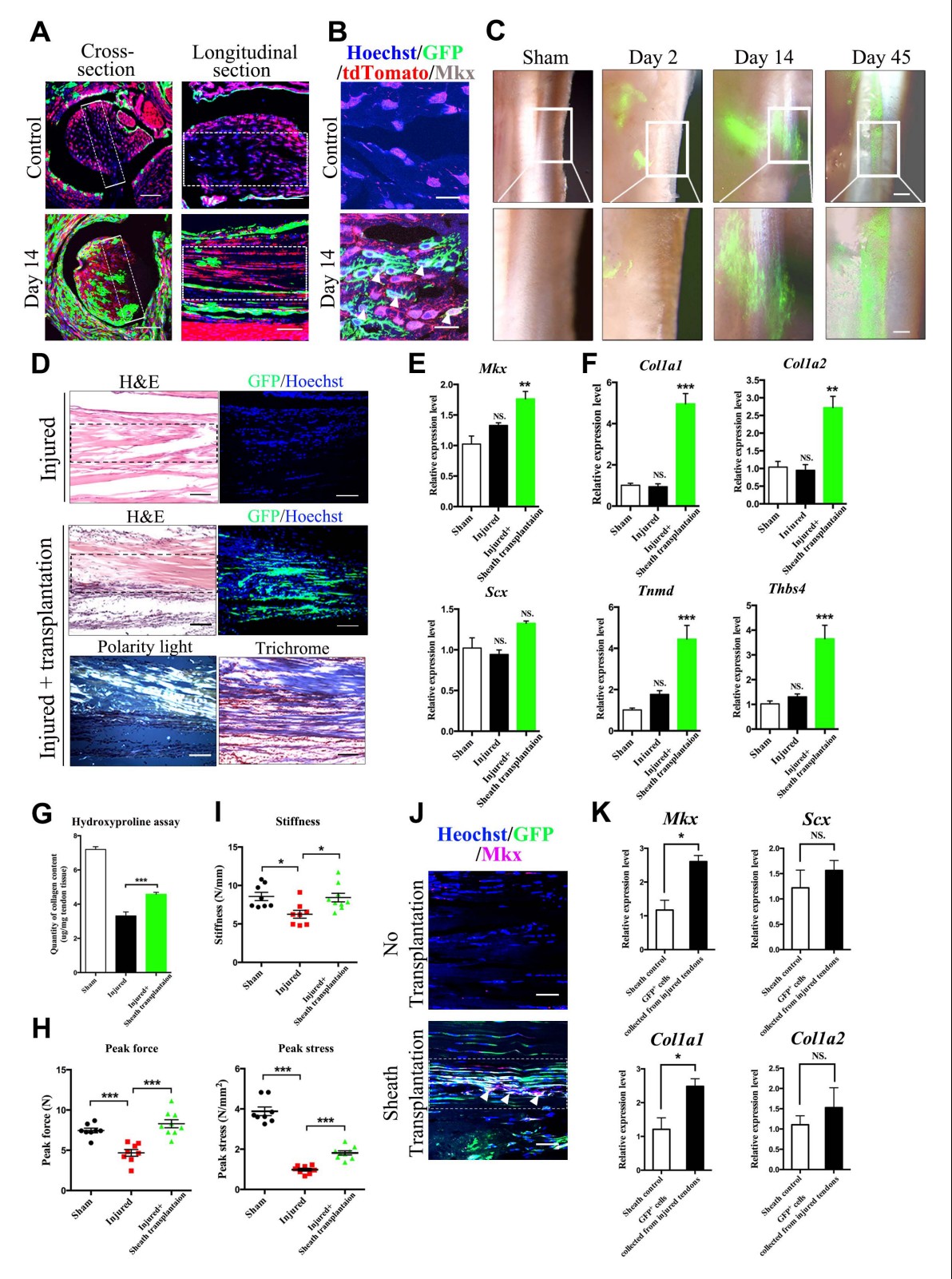

**Figure 3.** Tendon sheath tissues contribute to tendon repair. (**A**) Fluorescence analysis in cross- and longitudinal sections of the Tibialis anterior tendon of *BGLAP-Cre;Rosa26^{mT/mG}* mice at Day 0 (Control) and Day 14 after injury(without grafting) (n = 3 mice per group). Boxed areas indicate the regions of the injury sites (dashed line). Scale bar, 100 μm. (**B**) Fluorescence analysis of Mkx in the *BGLAP-Cre;Rosa26^{mT/mG}* mice Tibialis anterior tendon tissues at Day 0 (Control) and Day 14 after injury (n = 3 mice per group). White arrowheads point to GFP⁺/Mkx⁺ cells at the injured site of the *BGLAP-Cre;*
*Figure 3 continued on next page*

*Figure 3 continued*

*Rosa26^mT/mG* mouse tendon tissues. Scale bar, 20 μm. (**C**) GFP⁺ Tibialis anterior tendon sheath tissues isolated from the *BGLAP-Cre;Rosa26^mT/mG* mice formed tendon-like fibers at the injured Achilles tendon of immunocompromised mice after transplantation. Scale bar, 250 μm. Boxed regions of higher magnification are shown in the lower panel. Scale bar, 100 μm (n = 5 mice per group). (**D**) Histologic analysis of consecutive sections at the injured sites of the sheath transplantation group (bottom) and the no transplantation group (top) at Day 14 after injury (n = 5 mice). Boxed areas indicate the regions of the injury sites (dashed line). (**E, F**) Gene expression profiles of (**E**) tendon progenitor markers and (**F**) tendon extracellular matrix (ECM) components of the injured tendon at Day 14. Relative expression levels were normalized to *Gapdh* and the sham group. Data are means ± s.e.m and were analyzed using one-way analysis of variance (ANOVA) followed by Tukey's multiple comparison test. **p≤0.0021; ***p≤0.0008; NS, not significant. n = 4 mice per group. (**G**) Hydroxyproline assay of Achilles tendon tissues 4 weeks after injury. Data are means ± s.e.m and were analyzed using one-way analysis of variance (ANOVA) followed by Tukey's multiple comparison test. ***p≤0.0007. n = 5 mice per group. (**H, I**) Parameters of mechanical testing of tendon tissues in the sham, tendon-injured and tendon=injured with *BGLAP-Cre;Rosa26^mT/mG* sheath transplantation groups at 4 weeks after surgery. Sheath transplantation represents GFP⁺ sheath-derived cells sorted from the *BGLAP-Cre;Rosa26^mT/mG* mice. Data are means ± s.e.m and were analyzed using one-way analysis of variance (ANOVA) followed by Tukey's multiple comparison test. ***p≤0.001, *p≤0.0228. n ≥ 8 mice per group. (**J**) Co-localization of Mkx and BGLAP-Cre/GFP at the injured sites of the nude mice tendon tissues with or without the sheath transplantation at Day 14 after injury (n = 5 mice per group). White arrowheads point to GFP⁺/Mkx⁺ cells at the injured site of the nude mice tendon tissues with *BGLAP-Cre;Rosa26^mT/mG* sheath transplantation. Boxed areas indicate the regions of the injury sites (dashed line). Scale bar, 20 μm. (**K**) Gene-expression profiles of tendon progenitor markers and ECM components of FACS-sorted GFP⁺ cells from the nude mice injured tendons with *BGLAP-Cre;Rosa26^mT/mG* sheath transplantation at Day 14. Sheath controls were the FACS-sorted GFP⁺ cells directly from *BGLAP-Cre;Rosa26^mT/mG* sheath tissues. Relative expression levels were normalized to *β-tubulin* and the sheath control group. Data are means ± s.e.m and were analyzed using Student's t-tests. *p≤0.0383; NS, not significant. n = 3 replicates per group.

DOI: https://doi.org/10.7554/eLife.30474.015

The following source data and figure supplements are available for figure 3:

**Source data 1.** Source data relating to *Figure 3E*.
DOI: https://doi.org/10.7554/eLife.30474.018
**Source data 2.** Source data relating to *Figure 3F*.
DOI: https://doi.org/10.7554/eLife.30474.019
**Source data 3.** Source data relating to *Figure 3G*.
DOI: https://doi.org/10.7554/eLife.30474.020
**Source data 4.** Source data relating to *Figures 3H* and 3I.
DOI: https://doi.org/10.7554/eLife.30474.021
**Source data 5.** Source data relating to *Figure 3K*.
DOI: https://doi.org/10.7554/eLife.30474.022
**Figure supplement 1.** *BGLAP-Cre/GFP⁺cells* contribute to tendon repair in *BGLAP-Cre; Rosa26^mT/mG* mice.
DOI: https://doi.org/10.7554/eLife.30474.016
**Figure supplement 2.** GFP⁺sheath cells differentiate into Mkx-expressing cells and contribute to tendon repair.
DOI: https://doi.org/10.7554/eLife.30474.017

differentiation process in tendon progenitor cells. Thus, we examined the gene profiles of tendon markers in wild-type mice after injury (*Figure 4A–C*). We found that tendon progenitor markers and tendon ECM markers were all significantly upregulated after injury (*Figure 4A,B*). We also noticed that Hh ligand Shh, but not Dhh and Ihh, was significantly upregulated in the sheath tissues after injury (*Figure 4C*). Thus, we tested the Hh signaling activity after injury *in vivo* by using a *Ptch1^lacZ/+* mouse model, which is derived from a floxed allele of *Ptch1* (*Mak et al., 2006*). *Ptch1* is a transcriptional target of Hh signaling, and the expression of the *LacZ* gene represents the activity of Hh signaling. Conditional loss of *Ptch1* by crossing with *Stra8-Cre* mice leads to gain of *LacZ* expression in the germline embryowide. A full-thickness medial edge of the defect was introduced into the Tibialis anterior tendon of the left hind limb of the *Ptch1^lacZ/+* mice; the tendon fibers in this region were surrounded by true tendon sheaths. Interestingly, strong X-gal staining was observed at 2 weeks after injury in the surrounding sheath tissues (*Figure 4D,E*). Hh target gene *Gli1*, as well as the tendon-sheath markers *Tppp3* and *Bglap*, were strongly induced in the injured tissues (*Figure 4F*). To further demonstrate that cells with activated Hh signaling from the sheath tissues contribute to tendon repair, we isolated sheath tissues from the *Ptch1^lacZ/+* mice and transplanted them into immunocompromised mice with injuries in the Achilles tendon, where no true tendon sheaths were present (*Figure 4G*). Two weeks after transplantation, *X-gal* signals were found in the injured tendon fibers and *LacZ⁺* cells were detected within the injured regions (*Figure 4H*).

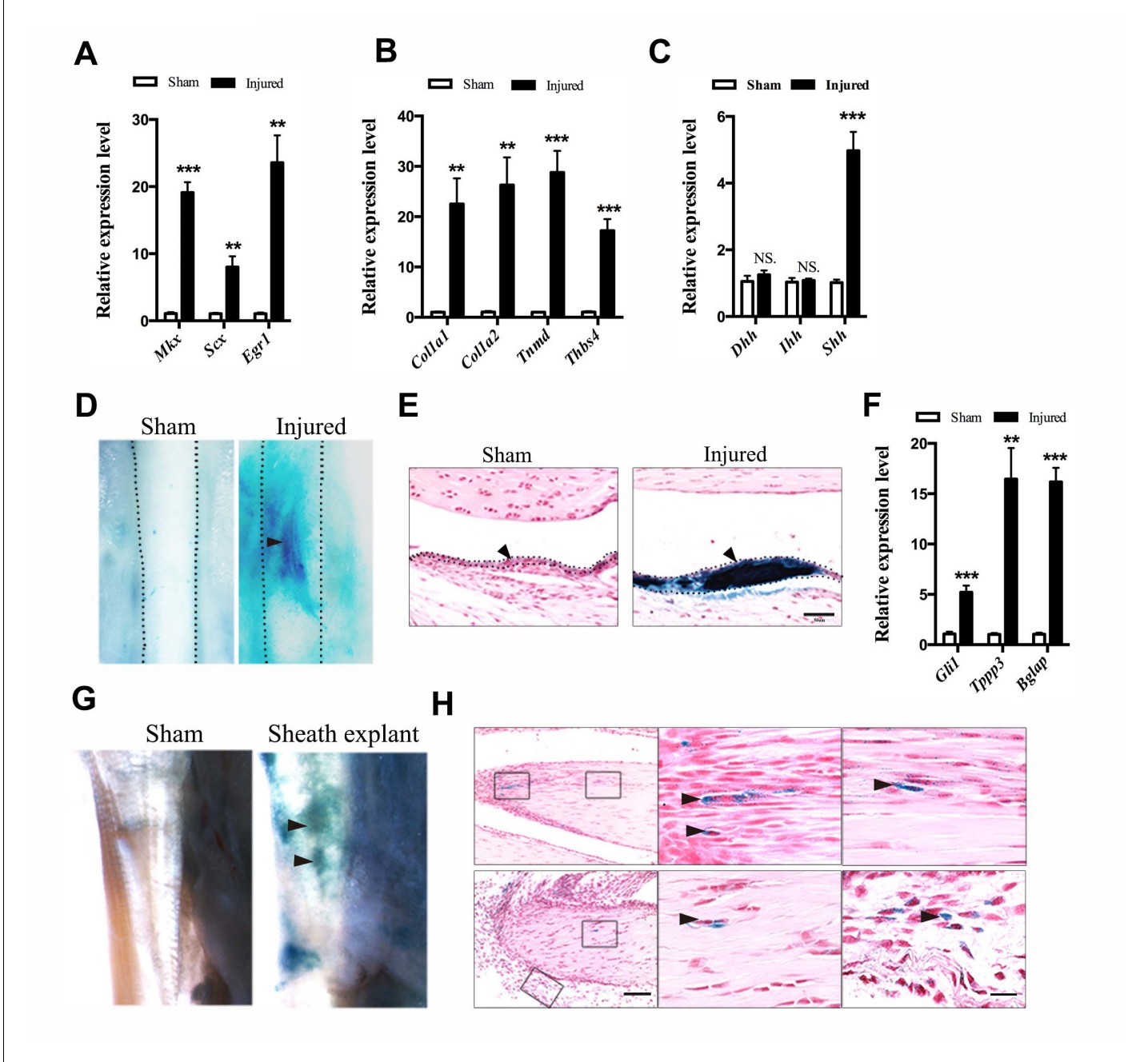

**Figure 4.** Activation of Hh signaling in sheath tissues upon injury. (A, B) Gene expression profiles of (A) tendon progenitor markers and (B) ECM components in the tendon sheath tissues of wild-type mice two weeks after injury. Relative expression levels were normalized to $\beta$-tubulin and the sham group. Data are means ± s.e.m and were analyzed using Student's t-tests. **p≤0.0055; ***p≤0.0006. n = 4 mice per group. (C) Gene expression profiles of Hh ligands in the sheath tissues one week after injury. Relative expression levels were normalized to *Gapdh* and the sham group. Data are means ± s. e.m and were analyzed using Student's t-tests. ***p≤0.0001; NS, not significant. n = 6 mice per group. (D) X-gal (USB Corporation, USA) staining of the Tibialis anterior tendon of *Ptch1LacZ/+* mice two weeks after injury. Strong signals were observed in the sheath tissues of the injured site (arrowhead). Dotted lines depict the boundaries of tendon tissues (n = 5 mice per group). (E) Cross-section of the injured sheath tissues as shown in (D) (n = 5 mice per group). Dotted lines depict the boundaries of the sheath tissues. Scale bar, 50 µm. (F) Gene expression profiles in the tendon sheath tissues of wild-type mice two weeks after injury. Relative expression levels were normalized to $\beta$-tubulin and the sham group. Data are means ± s.e.m and were analyzed using Student's t-tests. **p≤0.0025; ***p≤0.001. n = 4 mice per group. (G) *Ptch1LacZ/+* sheath tissues transplanted to the injured Achilles tendon of immunocompromised mice after two weeks (n = 5 mice per group). (H) Histologic analysis of the injured tendon fibers as shown in (G). Scale bar, 100 µm. Boxed areas are shown in high magnification on the right. The lower right panel depicts the boxed area from the surrounding tissues. Arrowheads point to *LacZ+* cells within the tendon fibers (n = 5 mice per group). Scale bar, 20 µm.

*Figure 4 continued on next page*

*Figure 4 continued*

DOI: https://doi.org/10.7554/eLife.30474.023

The following source data is available for figure 4:

**Source data 1.** Source data relating to *Figure 4A*.

DOI: https://doi.org/10.7554/eLife.30474.024

**Source data 2.** Source data relating to *Figure 4B*.

DOI: https://doi.org/10.7554/eLife.30474.025

**Source data 3.** Source data relating to *Figure 4C*.

DOI: https://doi.org/10.7554/eLife.30474.026

**Source data 4.** Source data relating to *Figure 4F*.

DOI: https://doi.org/10.7554/eLife.30474.027

To further elucidate the biological roles of Hh signaling in sheath tissues, we generated the *Ptch1$^{c/c}$;BGLAP-Cre* mutant mice to upregulate Hh signaling in the tendon sheath tissues. Histologic examination revealed that the sheath tissues in the mutant mice were significantly thicker than those of the control littermates (*Figure 5A,B*). These data suggest that the sheath cells were highly proliferative. As expected, strong X-gal signals were detected in the sheath tissues of the *Ptch1$^{c/c}$;BGLAP-Cre* mutant mice, indicating that Hh signaling activity was strongly induced (*Figure 5C–E*). Indeed, the thickened tendon sheaths of the *Ptch1$^{c/c}$;BGLAP-Cre* mutant mice showed strong expression of phospho-H3 and there were very few apoptotic cells (*Figure 5F*). We also found that tendon sheath tissues were normally more proliferative than tendon fiber tissues in adult tissues (*Figure 5—figure supplement 1A*). Strikingly, the highly proliferative sheath cells also co-expressed Gli and Mkx in the sheath tissues (*Figure 5G,H*), suggesting that Hh signaling promotes the proliferation of tendon sheath progenitor cells. Indeed, Mkx has been previously shown to be strongly expressed in the sheath tissues (*Liu et al., 2010*) and implicated in both tendon repair (*Liu et al., 2015*) and type I collagen production in tendon tissues (*Ito et al., 2010*). Similarly, primary tendon sheath cells isolated from the *Ptch1$^{c/c}$;BGLAP-Cre* mutant mice also displayed positive X-gal signals and increased cell proliferation (*Figure 5—figure supplement 1B–F*). Together, our data indicate that Hh signaling promotes the proliferation of *Mkx*-expressing sheath stem/progenitor cells.

## Hh signaling in tendon sheath is required for tendon repair

Next, to test whether Hh signaling in the tendon sheath is required for tendon repair, we genetically removed Hh signaling by generating *Smo$^{c/c}$;BGLAP-Cre* mutant mice. In contrast to those of the *Ptch1$^{c/c}$;BGLAP-Cre* mutant mice, the tendon sheaths of the *Smo$^{c/c}$;BGLAP-Cre* mutant mice were much thinner than those of control littermates (*Figure 6A,B*). Sheath cell proliferation was significantly reduced while the frequency of apoptotic cells was slightly increased (*Figure 6C*). Mkx and Gli1 expression were minimally expressed in normal but not in mutant sheath tissues (*Figure 6D*).

Under tendon-injured conditions, the tendon sheath tissues around the injured site in the control group were markedly thickened (*Figure 7A*). Both Gli1 and Mkx were strongly expressed in the sheath tissues (*Figure 7B*). These observations were highly similar to those in the *Ptch1$^{c/c}$;BGLAP-Cre* mutant mice. However, these phenotypic changes were not observed in the *Smo$^{c/c}$;BGLAP-Cre* mutant mice upon injury (*Figure 7A,B*). Tendon repair was significantly impaired in the *Smo$^{c/c}$;BGLAP-Cre* mutant mice and cell proliferation was inhibited in the injured tendon fibers of these mice (*Figure 7C,D*). Total collagen content was significantly reduced (*Figure 7E*) and co-expression of Mkx and Gli1 in the mutant tendon fibers was nearly missing (*Figure 7F*). Gene expression levels of *Mkx*, *Scx* and all the tendon ECM markers were similarly reduced (*Figure 7—figure supplement 1*). In summary, our data indicate that Hh signaling is necessary to promote the proliferation of sheath progenitor cells for tendon repair.

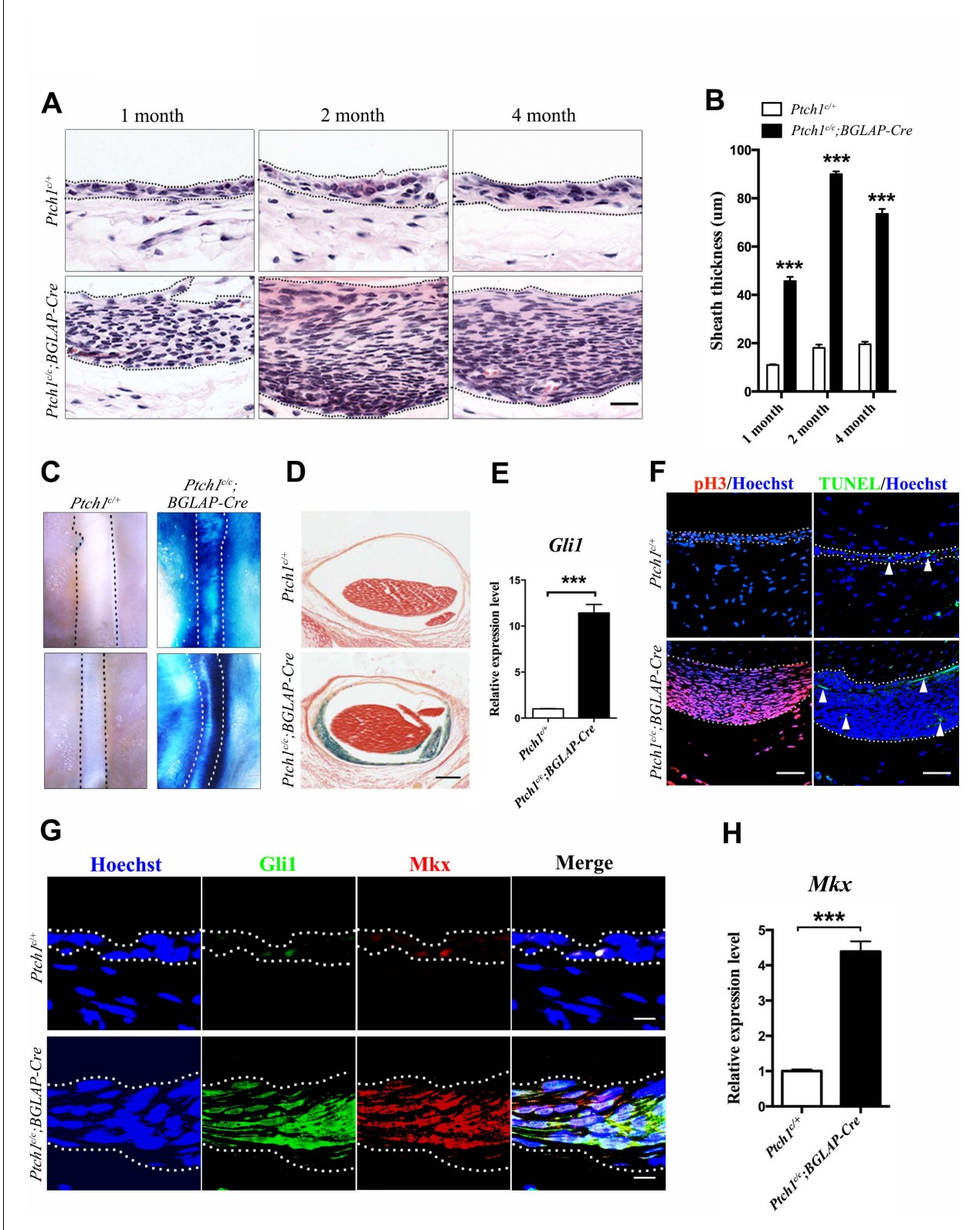

**Figure 5.** Activated Hh signaling increases cell proliferation in sheath tissues. (**A**) Histologic analysis of the sheath tissues of the Tibialis anterior tendon of different mice genotypes at multiple time points. Scale bar, 20 μm. (**B**) Statistical analyses of the sheath thickness data illustrated in (**A**). Data are means ± s.e.m and were analyzed using Student's t-tests. ***p<0.0001. n = 10 biological replicates per group. (**C**) X-gal staining of the sheath tissues of the Tibialis anterior tendon (upper panel) and the Peroneus longus tendon (lower panel) of mice of different genotypes at 3 months old (n = 6 mice per

*Figure 5 continued on next page*

*Figure 5 continued*

group). (D) Cross-sections of the Tibialis anterior tendon sheath tissues at Day 21 with X-gal staining (n = 6 mice per group). Scale bar, 100 μm. (E) QRT-PCR analysis of Hh activity in the sheath tissues. Relative expression levels were normalized to *Gapdh* and the *Ptch1$^{c/+}$* group. Data are means ± s.e.m and were analyzed using a Student's t-test. ***p≤0.0004. n = 3 mice per group. (F) Fluorescent immunohistochemistry of cell proliferation (pH3) and apoptosis (TUNEL) of the sheath tissues at 2 months old of mice with genotypes as shown (n = 3 mice per group). White arrowheads point to apoptotic cells in the sheath tissues. Scale bar, 40 μm. (G) Co-localization of Gli1 and Mkx expression in the sheath tissues of mice with genotypes as shown (n = 3 mice per group). Scale bar, 10 μm. (H) Gene expression analysis of *Mkx* in the sheath tissues illustrated in (G). Relative expression levels were normalized to *Gapdh* and the *Ptch1$^{c/+}$* group. Data are means ± s.e.m and were analyzed using a Student's t-test. ***p≤0.0007. n = 3 mice per group. Dotted lines depict the boundaries of sheath tissues. Additional data for this figure are provided in *Figure 5—figure supplement 1*.
DOI: https://doi.org/10.7554/eLife.30474.028

The following source data and figure supplements are available for figure 5:

**Source data 1.** Source data relating to *Figure 5B*.
DOI: https://doi.org/10.7554/eLife.30474.031
**Source data 2.** Source data relating to *Figure 5E*.
DOI: https://doi.org/10.7554/eLife.30474.032
**Source data 3.** Source data relating to *Figure 5H*.
DOI: https://doi.org/10.7554/eLife.30474.033
**Figure supplement 1.** Hh signaling increases sheath cell proliferation.
DOI: https://doi.org/10.7554/eLife.30474.029
**Figure supplement 1—source data 1.** Source data relating to *Figure 5—figure supplement 1F*.
DOI: https://doi.org/10.7554/eLife.30474.030

## Hh signaling induces Mkx and Collagen I expression through TGFβ/Smad3 signaling

To dissect the mechanism through which Hh activation promotes the proliferation of tendon progenitor cells for tendon repair, we first infected primary *Ptch1$^{c/c}$* sheath cells with Cre adenovirus (*Figure 8A*), or treated GFP$^+$ tendon sheath cells from *BGLAP-Cre;Rosa26$^{mT/mG}$* mice with the Hh agonist purmorphamine (*Figure 8B*). Both sets of results showed significant upregulation of gene expression for sheath, tendon progenitor and ECM markers. Interestingly, the expression of *TGF-β1*, *-β2*, *-β3* and the TGFβ-signaling target gene *Smad7* were also greatly induced (*Figure 8C,D*), suggesting that Hh signaling activates TGFβ signaling in tendon sheath cells. Furthermore, Smad3 was also strongly phosphorylated in *Ptch1$^{c/c}$;BGLAP-Cre* mutant sheath tissues (*Figure 8E*). Next, we searched for *Smad*-binding sites in the promoter regions of *Mkx*. As expected, we found putative *Smad3* binding sites at positions −4060 bp,−714 bp and −156 bp of the *Mkx* promoter regions (*Figure 9A*). We cloned the promoter regions containing the putative binding sites and examined *Smad* binding using a luciferase assay. Our results showed that only *Smad3* but not *Smad2* significantly induced luciferase activities in both the 0.8 kb and 4.1 kb promoter regions of *Mkx*, indicating that *Smad3* is responsible for the induction of *Mkx* expression. To demonstrate the specificity of the Hh-induced activation of TGFβ signaling in inducing *Mkx* and *Col1a1* expression, we co-treated the primary sheath cells with purmorphamine and a specific inhibitor of the type I TGFβ receptor SB431542 (*Figure 9B,C*). In the presence of the SB431542 inhibitor, the effects of Hh agonists were greatly diminished. Together, our data indicate that TGFβ signaling mediates the effect of Hh signaling in regulating *Mkx* and *Col1a1* expression.

To elucidate the relevance of Hh signaling activation in human tendon injury, we also examined the expression of GLI1 and MKX in human tendinopathy specimens (*Figure 10*). We found that only regions with tendinopathy showed strong expression of GLI1 and MKX. These proteins were expressed at low levels in adjacent tendon regions where the integrity of the tendon fibers was relatively normal. Thus, our data suggest that Hh signaling is activated in human tendinopathy patients. However, it remains unclear whether GLI1/MKX expression represents a scarring and/or tendon repair process.

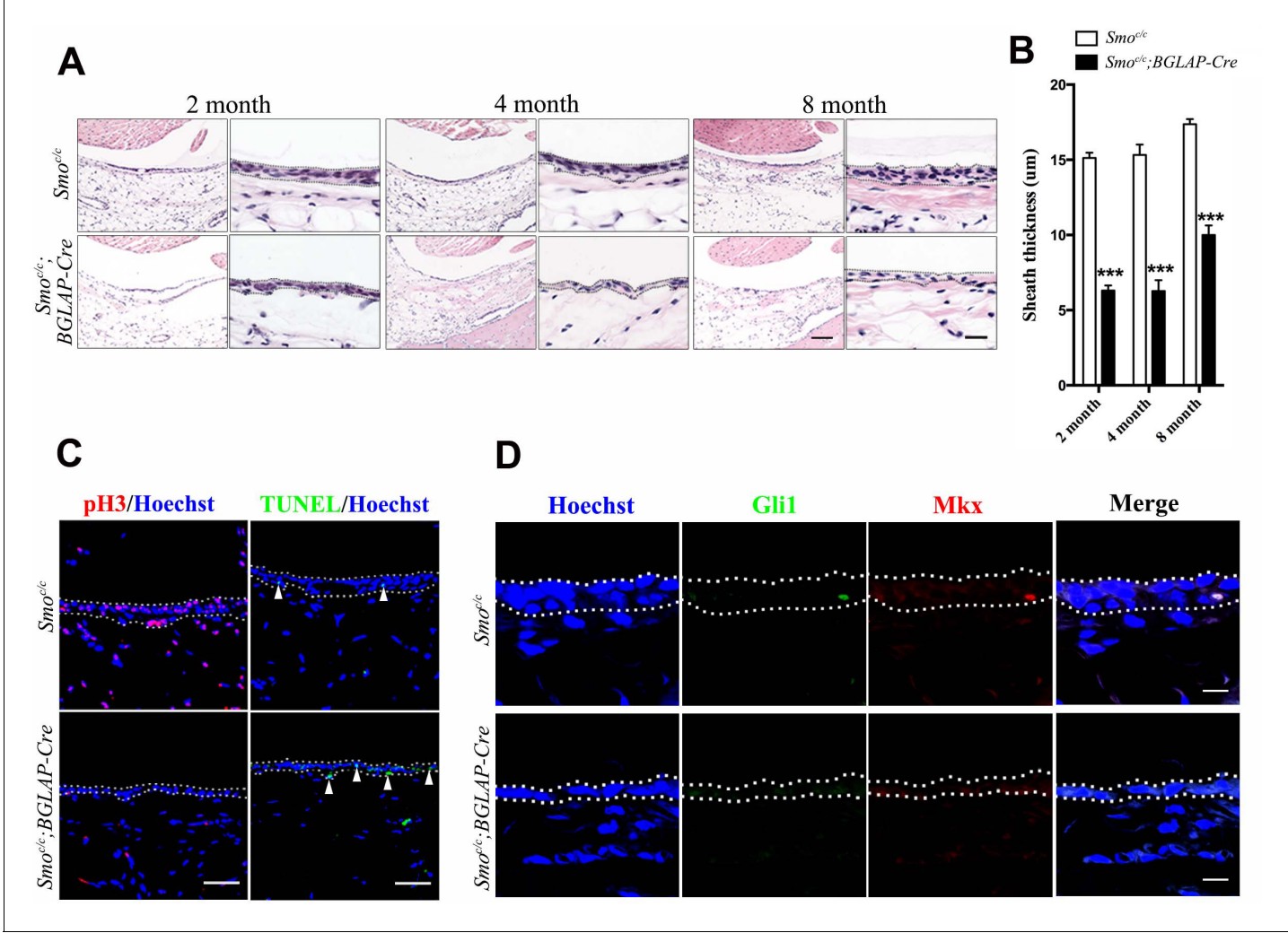

**Figure 6.** Hh signaling is necessary for sheath cell proliferation. (A) Histologic analysis of the sheath tissues of the Tibialis anterior tendon of mice with genotypes as shown. Scale bar, 100 µm. Higher magnification is shown on the right. Scale bar, 20 µm. (B) Statistical analysis of sheath thickness as shown in (A). Data are means ± s.e.m and were analyzed using Student's t-tests. ***p<0.0001. n = 10 biological replicates per group. (C) Fluorescent immunohistochemistry of sheath tissues showing the cell proliferation (pH3) and apoptosis (TUNEL) of mice with genotypes as shown. White arrowheads point to apoptotic cells in sheath tissues (n = 3 mice per group). Scale bar, 40 µm. (D) Immunohistochemistry of Gli1 and Mkx expression in the sheath tissues of mice with genotypes as shown (n = 3 mice per group). Scale bar, 10 µm. Dotted lines depict the boundaries of sheath tissues.
DOI: https://doi.org/10.7554/eLife.30474.034

The following source data is available for figure 6:

**Source data 1.** Source data relating to *Figure 6B*.
DOI: https://doi.org/10.7554/eLife.30474.035

## Discussion

To delineate the contribution of extrinsic sheath tissues in tendon repair, we need to identify specific adult tendon sheath markers for molecular and cellular tracing during the healing process. Our findings indicate that Osteocalcin (*Bglap*) is specifically expressed in adult tendon sheath tissues but not in tendon fibers, and therefore is an ideal sheath tissue marker. Previously, Bglap was thought to be expressed only in mature osteoblasts and odontoblasts, contributing to the composition of the extracellular matrix (ECM) of the bones (*Butler et al., 1992*; *Ducy et al., 1996*). More recently, Bglap was also found to be a secreted protein that binds to its receptor GPRC6A and shows broad paracrine effects on pancreatic islets, fat, muscle, brain and testes in

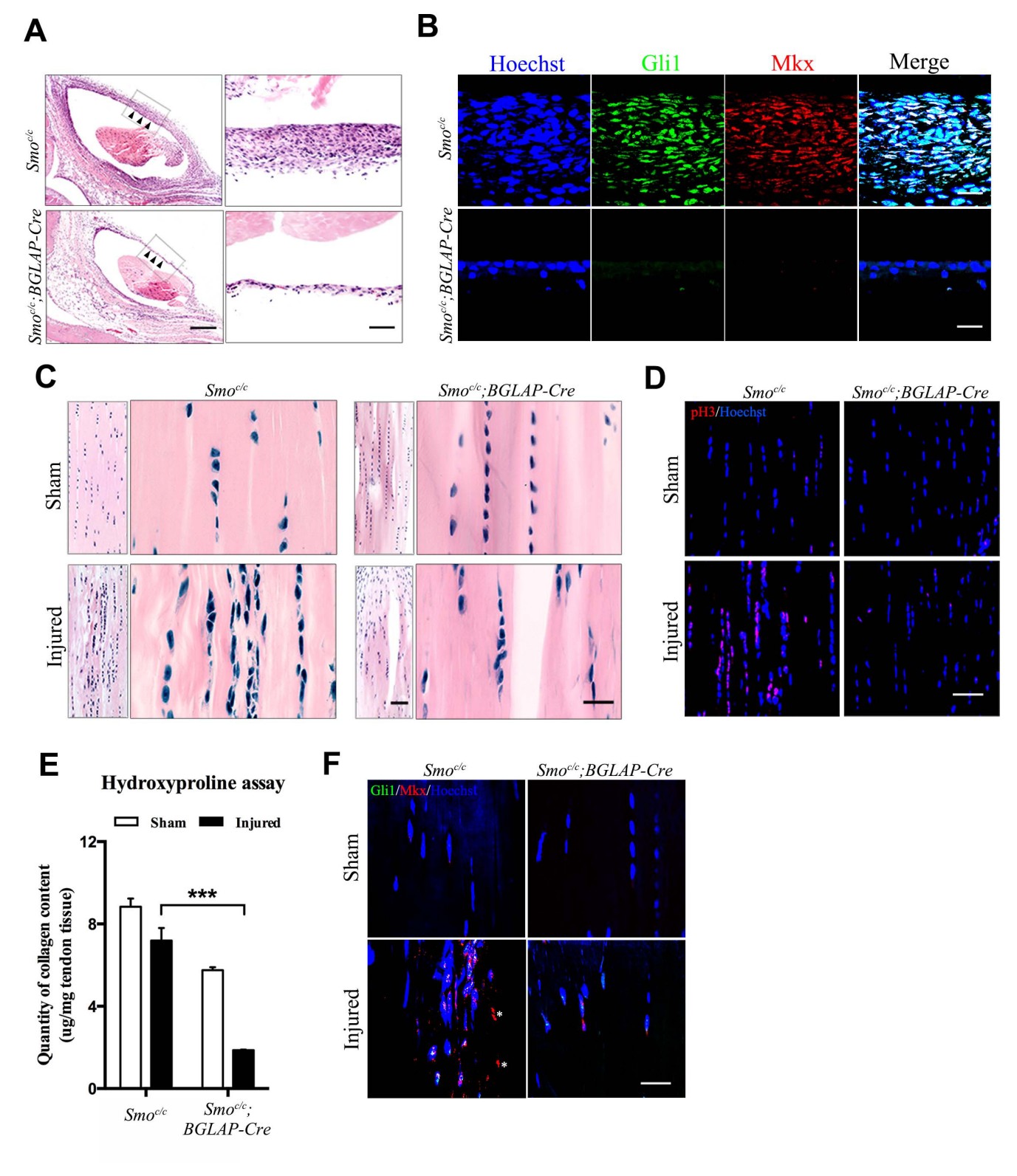

**Figure 7.** Hh signaling is required for sheath-mediated tendon repair. (**A**) Histologic analysis of the Tibialis anterior tendon of mice with genotypes as shown 4 weeks after injury (n = 4 mice per group). Scale bar, 400 μm. Higher magnifications of the boxed regions are shown on the right. Scale bar, 50 μm. Arrowheads point to the sheath tissues. (**B**) Fluorescent immunohistochemistry of Gli1 and Mkx expression in the sheath tissues as shown in (**A**) (n = 4 mice per group). Scale bar, 20 μm. (**C**) Histologic analysis of Tibialis anterior tendon fibers of mice with genotypes as shown 4 weeks after injury.
*Figure 7 continued on next page*

*Figure 7 continued*

Scale bar, 50 μm. Higher magnification is shown on the right (n = 4 mice per group). Scale bar, 20 μm. (**D**) Fluorescent immunohistochemistry of cell proliferation (pH3) of the tendons of mice with genotypes as shown in (**C**) (n = 4 mice per group). Scale bar, 40 μm. (**E**) Hydroxyproline assay using Tibialis anterior tendon tissues 4 weeks after injury. Data are means ± s.e.m and were analyzed using one-way analysis of variance (ANOVA) followed by Tukey's multiple comparison test. ***p≤0.0001. n = 4 mice per group. (**F**) Fluorescent immunohistochemistry of Gli1 and Mkx of the Tibialis anterior tendon fibers of mice with genotypes as shown 4 weeks after injury (n = 4 mice per group). *, non-specific signals. Scale bar, 20 μm. Additional data for this figure are provided in *Figure 7—figure supplement 1*.

DOI: https://doi.org/10.7554/eLife.30474.036

The following source data and figure supplements are available for figure 7:

**Source data 1.** Source data relating to *Figure 7E*.
DOI: https://doi.org/10.7554/eLife.30474.040
**Figure supplement 1.** Tendon repair is significantly impaired in the *Smo^c/c;BGLAP-Cre* mutant mice after injury.
DOI: https://doi.org/10.7554/eLife.30474.037
**Figure supplement 1—source data 1.** Source data relating to *Figure 7—figure supplement 1A*.
DOI: https://doi.org/10.7554/eLife.30474.038
**Figure supplement 1—source data 2.** Source data relating to *Figure 7—figure supplement 1B*.
DOI: https://doi.org/10.7554/eLife.30474.039

mice (*Ferron et al., 2008*; *Oury et al., 2011, 2013*; *Wei et al., 2014*; *Mera et al., 2016*). Although the function of *Bglap* in sheath tissues remains to be determined, it is conceivable that it serves at least in part as an ECM component of the sheath tissues and potentially regulates the metabolism of local tissues that contribute to tendon repair via autocrine or paracrine mechanisms.

Our studies showed that extrinsic sheath tissues possess stem/progenitor cells and contribute to tendon repair. It remains to be determined, however, whether these sheath-derived *Bglap*+ lineage cells differentiate into genuine tenocytes or just scarring sheath cells that invade into the tendon proper after tendon injury. Our data show that the co-expression of *Mkx* and *Col1a1* in the *Bglap*+-cells is indistinguishable between tenocytes and sheath cells. More specific markers may be required to address this question definitively.

Interestingly, true tendon sheath does not continuously envelope the tendon fibers entirely. It covers only areas that are subjected to high levels of fiction, and is believed to lubricate the surrounding tissues during movement. Our findings unravel an additional novel function of sheath tissues that is specific for tendon remodeling and repair. As tendon fibers in regions of high fiction are more vulnerable to wearing and injuries, the proximity of progenitor cells in the sheath tissues provides an efficient source of cells for local or short-range migration for tendon fiber repair.

Our findings demonstrate that Hh signaling is strongly activated upon injury, possibly due to the strong induction and secretion of Shh (*Figure 4C*). In fact, activation of Hh signaling in sheath tissues triggers the proliferation of sheath progenitor cells. This generates a pool of progenitor cells for tendon differentiation and tendon fiber repair. It has been shown previously that Hh signaling regulates the early differentiation of progenitor cells at the tendon insertion site during embryonic development (*Liu et al., 2012, 2013*). Our results further reveal the importance of activation of Hh signaling in early events during adult tendon repair. In addition, the activation of Hh signaling results in significant upregulation of *Mkx* expression. *Mkx* plays an important role in the differentiation of tendon progenitor cells (*Ito et al., 2010*; *Liu et al., 2010*; *Suzuki et al., 2016*). It has been shown that *Mkx*-expressing mesenchymal stem cells (MSCs) significantly promote tendon healing by activation of TGFβ signaling (*Liu et al., 2015*). Our results also show that Hh signaling activates TGFβ signaling. TGFβ signaling is a major regulatory pathway for tenogenesis. It is essential for the recruitment and maintenance of tendon progenitors for tendon formation (*Pryce et al., 2009*). It also induces collagen formation and matrix remodeling during tendon healing (*Hou et al., 2009*; *Lu et al., 2017*; *Potter et al., 2017*; *Kayama et al., 2016*). Thus, activation of Hh signaling plays at least two roles. First, as an early event after injury, it ensures the availability of a sufficient pool of progenitor cells for repair. Second, it activates signaling cues for further cell differentiation and ECM production for tendon fiber regeneration. Our work suggests that activation of Hh signaling in sheath tissues may be a therapeutic intervention for tendon injury.

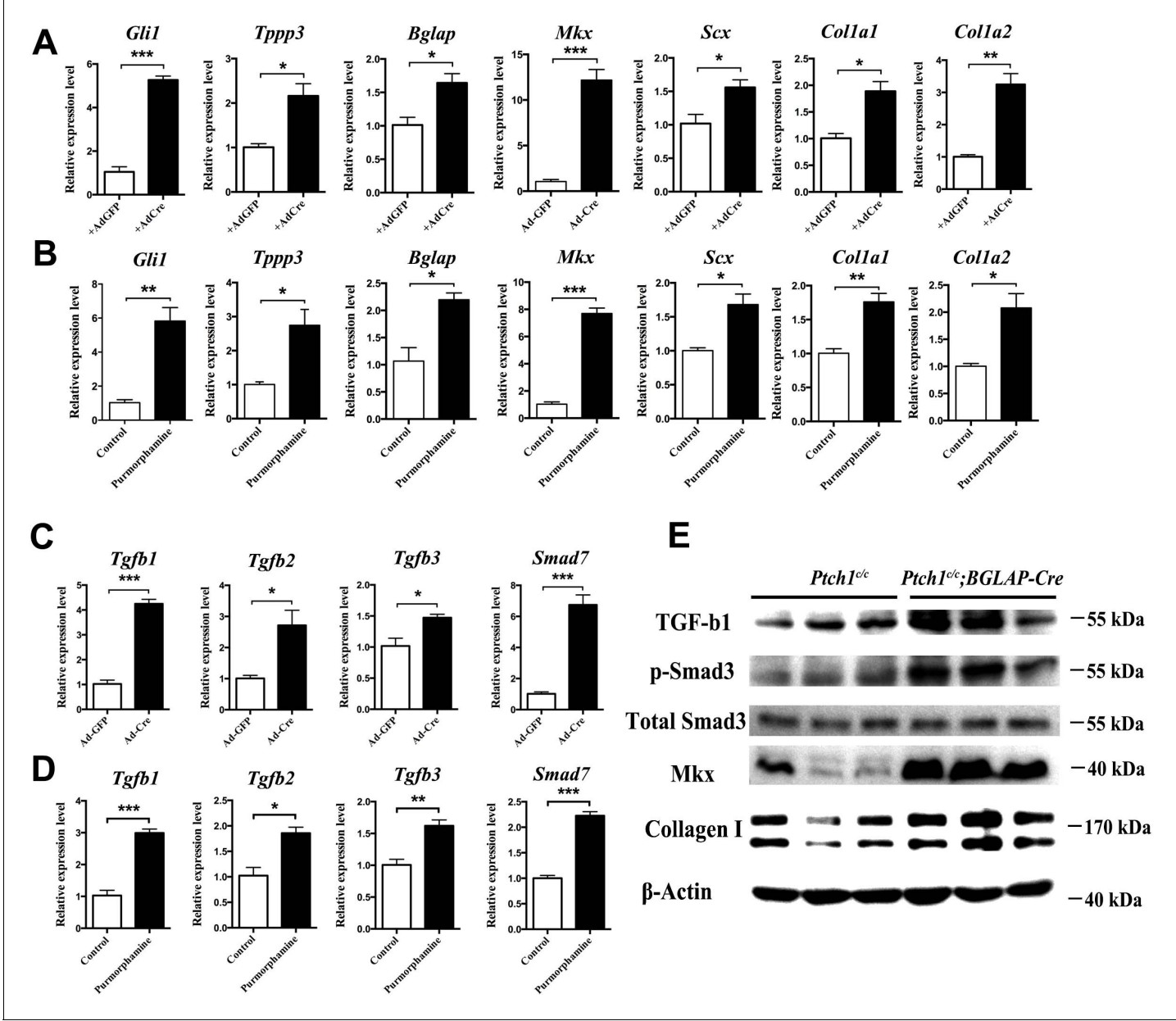

**Figure 8.** Hh activation upregulates Mkx and Collagen I via TGFβ/Smad3 signaling. (**A, C**) Gene expression profiling using primary *Ptch1^{c/c}* sheath cells infected with Cre-adenovirus for 48 hr. Relative expression levels were normalized to *β-tubulin* and the Ad-GFP group. Data are means ± s.e.m and were analyzed using Student's t-tests. *p≤0.0394; **p≤0.0028; ***p≤0.0009. n = 3 biological replicates per group. (**B, D**) Gene expression profiling using *BGLAP-Cre;Rosa26^{mT/mG}* sorted GFP⁺ primary sheath cells treated with the Hh agonist purmorphamine (1000 nM, Millipore, USA) for 48 hr. Relative expression levels were normalized to *β-tubulin* and the control group. Data are means ± s.e.m and were analyzed using Student's t-tests. *p≤0.0213; **p≤0.0085; ***p≤0.0007. n = 3 biological replicates per group. (**E**) Western blot analysis of protein expression in primary sheath cells of *Ptch1^{c/c}* and *Ptch1^{c/c};BGLAP-Cre* mice (n = 3 per group).

DOI: https://doi.org/10.7554/eLife.30474.041

The following source data is available for figure 8:

**Source data 1.** Source data relating to *Figure 8A*.
DOI: https://doi.org/10.7554/eLife.30474.042
**Source data 2.** Source data relating to *Figure 8B*.
DOI: https://doi.org/10.7554/eLife.30474.043
**Source data 3.** Source data relating to *Figure 8C*.
DOI: https://doi.org/10.7554/eLife.30474.044
**Source data 4.** Source data relating to *Figure 8D*.
DOI: https://doi.org/10.7554/eLife.30474.045

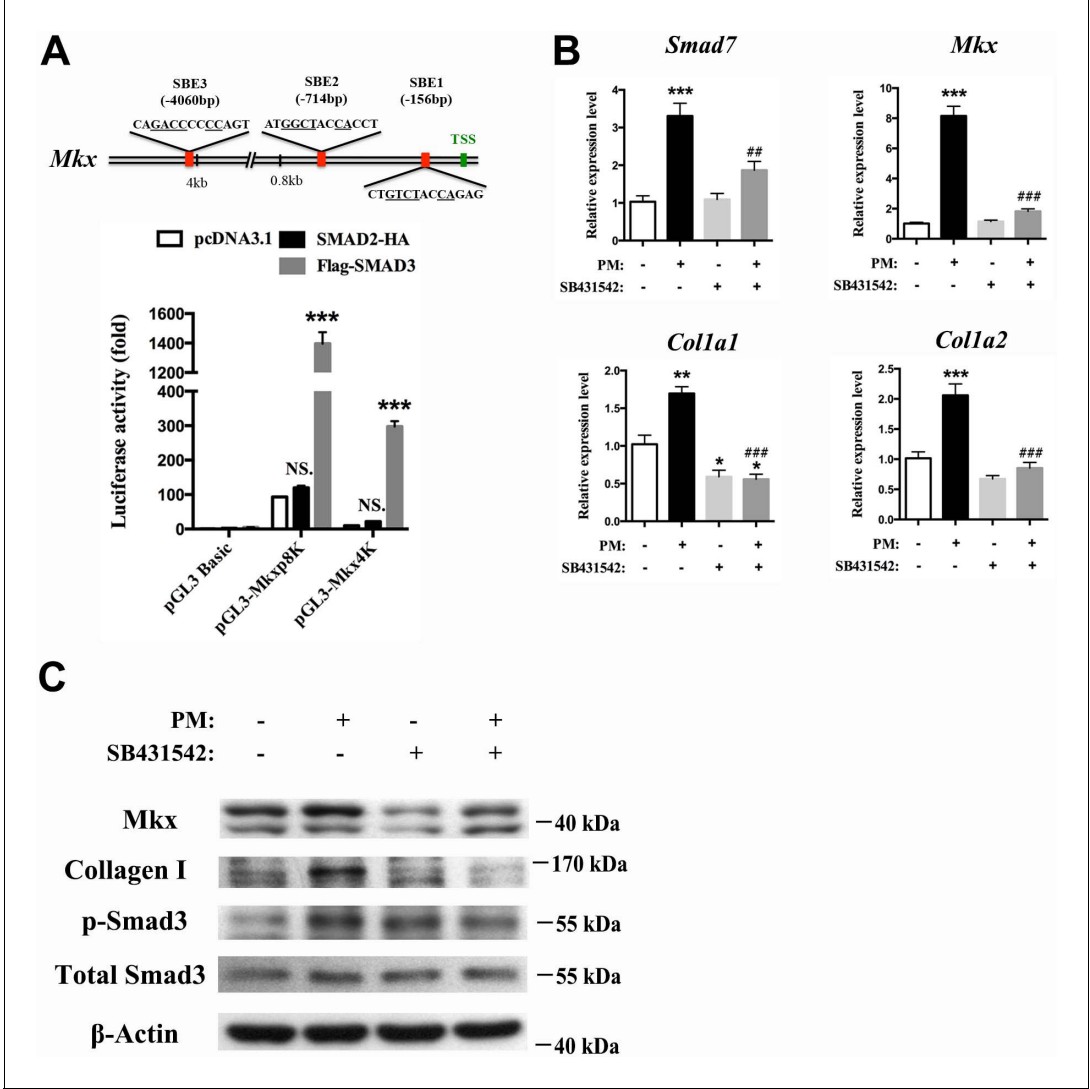

**Figure 9.** Hh activation upregulates Mkx and Collagen I via TGFβ/Smad3 signaling. (**A**) (Upper panel) Schematic diagram of putative *Smad*-binding element (SBE) motifs in the promoter region of *Mkx*. (TSS, transcription start site; the canonical *Smad*-binding conserved sequence GT/GCTNNCA is underlined). (Lower panel) Dual-luciferase assay of *Mkx* activities using reporter plasmids with 0.8 kb (*pGL3-Mkxp8K* with two *Smad* binding sites) or 4060 bp (*pGL3-Mkx4K* with three *Smad* binding sites) of *Mkx* promoter regions with *SMAD2* or *SMAD3* overexpression, respectively, in HEK293T cells. Data are means ± s.e.m and were analyzed using two-way analysis of variance (ANOVA) followed by Tukey's multiple comparison test. ***p≤0.0001; NS, not significant. n = 3 biological replicates per group. (**B**) QRT-PCR analysis of the TGFβ/Smad3 signaling target gene *Smad7*, the tendon progenitor *Mkx* and the main ECM components *Col1a1* and *Col1a2* in primary sheath cells treated with purmorphamine (PM) (1000 nM), with or without specific type I TGFβ receptor inhibitor SB431542 pre-treatment (10 μM, 1 hr). Relative expression levels were normalized to *β-tubulin* and the control group. Data are means ± s.e.m and were analyzed using one-way analysis of variance (ANOVA) followed by Tukey's multiple comparison test. *p≤0.0293; **p≤0.0012; ***p≤0.0003; ## p≤0.0045; ### p≤0.0001. (*, comparison with control group; #, comparison with PM treatment). n = 3 biological replicates per group. (**C**) Western blot analysis of protein expression in sheath primary cells (n = 3 per group).

DOI: https://doi.org/10.7554/eLife.30474.046

The following source data is available for figure 9:

**Source data 1.** Source data relating to *Figure 9A*.
DOI: https://doi.org/10.7554/eLife.30474.047

**Source data 2.** Source data relating to *Figure 9B*.
DOI: https://doi.org/10.7554/eLife.30474.048

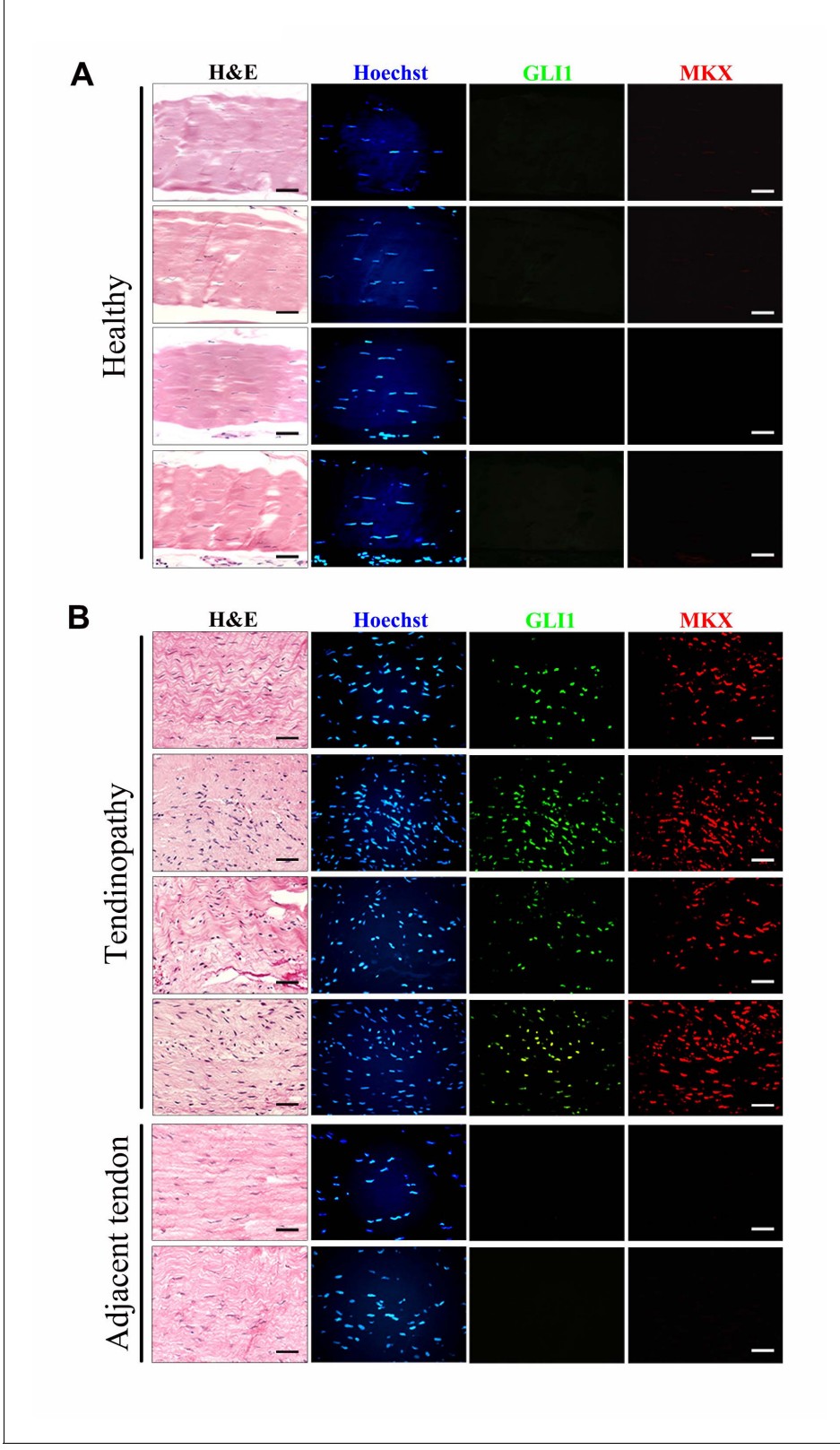

**Figure 10.** Upregulation of GLI1 and MKX expression in human tendinopathy specimens. (A, B) Fluorescent immunohistochemistry of GLI1 and MKX in (A) healthy patellar tendon tissues or (B) patellar tendinopathy tissues. The bottom images show two regions adjacent to the injured tissues and are representative of the tendon tissues of 'tendinopathy' patients. Different rows indicate samples from different healthy subjects or patients. (n = 7 healthy subjects or patients per group.)

*Figure 10 continued on next page*

*Figure 10 continued*

DOI: https://doi.org/10.7554/eLife.30474.049

In conclusion, we have identified *Bglap* as a new marker for adult sheath tissues. *Bglap*-expressing sheath cells possess stem cell properties and contribute to tendon repair in which Hh signaling is necessary and sufficient to promote the proliferation of *Mkx*[+] cells. Our results reveal important molecular evidence that Hh signaling is required for tendon repair that is mediated by extrinsic tendon sheath tissue.

# Materials and methods

## Key resources table

| Reagent type (species) or resource | Designation | Source or reference | Identifiers |
|---|---|---|---|
| Genetic reagent (*Mus musculus*) | *BGLAP-Cre;Rosa26*[mT/mG] | This paper | NA |
| Genetic reagent (*M. musculus*) | *Ptch1*[c/c]*;BGLAP-Cre* | *Mak et al. (2006)* | PMID: 16936073; DOI: 10.1242/dev.02546 RRID:MGI:3687744 |
| Genetic reagent (*M. musculus*) | *Smo*[c/c]*;BGLAP-Cre* | *Mak et al. (2008a)* | PMID: 18434416; DOI: 10.1242/dev.018044 RRID:MGI:5311416 |
| Genetic reagent (*M. musculus*) | *Ptch1*[LacZ/+] | This paper | NA |
| Recombinant DNA reagent | pGL3-Mkxp8K | This paper | NA |
| Recombinant DNA reagent | pGL3-Mkx4K | This paper | NA |
| Recombinant DNA reagent | pCMV5 SMAD2-HA | Addgene, USA | plasmid # 14930 |
| Recombinant DNA reagent | pCMV5B-Flag-SMAD3 | Addgene, USA | plasmid # 11742 |
| Antibody | Osteolcalcin/Bglap | Abcam, USA | ab93876, 5 ug/ml RRID: AB_10675660 |
| Antibody | Tppp3 | Abcam, USA | ab150998, 1:50 RRID: AB_2716739 |
| Antibody | Mkx | Lifespan Biosciences Inc., USA | LS-B8063, 1 ug/ml RRID: AB_2716740 |
| Antibody | Mkx | Abcam, USA | ab179597, 1 ug/ml RRID: AB_2716741 |
| Antibody | Gli1 | R&D, USA | MAB3324, 10 ug/ml RRID:AB_2111775 |
| Antibody | Gli2 | R&D, USA | AF3635, 5 ug/ml RRID:AB_2111902 |
| Antibody | pH3 | SantaCruz, USA | sc-8656-R, 1:200 RRID:AB_653256 |
| Antibody | Collagen I | Abcam, USA | ab292, 1:1000 RRID:AB_303415 |
| Antibody | TGF-b1 | Abcam, USA | ab64715, 1:500 RRID:AB_1144265 |
| Antibody | Smad3 | CST, USA | #9523, 1:1000 RRID:AB_2193182 |
| Antibody | p-Smad3 | CST, USA | #9520, 1:1000 RRID:AB_2193207 |
| Antibody | β-Actin | CST, USA | #8457, 1:10000 RRID:AB_10950489 |
| Commercial assay or kit | Apoptosis Fluorescein Detection Kit | Millipore, USA | s7111 |

*Continued on next page*

*Continued*

| Reagent type (species) or resource | Designation | Source or reference | Identifiers |
|---|---|---|---|
| Commercial assay or kit | Hydroxyproline Assay Kit | Sigma-Aldrich, USA | MAK008 |
| Commercial assay or kit | Dual-luciferase reporter assay system | Promega, USA | E1980 |
| Chemical compound, drug | BMP2 | CST, USA | #4697 |
| Chemical compound, drug | TGF-β1 | Peprotech, USA | 100–21 |
| Chemical compound, drug | Purmorphamine | Millipore, USA | 540220 |
| Chemical compound, drug | SB431542 | Sigma-Aldrich, USA | S4317 |

## Mice

The human *BGLAP-Cre* mice (*Zhang et al., 2002*) were crossed with the *Rosa26$^{mT/mG}$* mouse line (*Muzumdar et al., 2007*), *Ptch1$^{c/c}$* mice (*Mak et al., 2006*) or *Smo$^{c/c}$* mice (*Long et al., 2001*) to generate the *BGLAP-Cre;Rosa26$^{mT/mG}$* mice, *Ptch1$^{c/c}$;BGLAP-Cre* mice and *Smo$^{c/c}$;BGLAP-Cre* mice. The *Ptch1$^{c/c}$* mice were crossed with the *Stra8-Cre* mice (*Sadate-Ngatchou et al., 2008*) to generate *Ptch1$^{LacZ/+}$* mice. All experiments were approved by the Animal Experimentation Ethics Committee of the Chinese University of Hong Kong (16-637).

## Tendon injury mouse model

Tendon injury mouse models were generated as previously described (*Guerquin et al., 2013*). Briefly, 8–12 week-old male wild-type mice (C57BL/6), *Ptch1$^{LacZ/+}$* mice, *Smo$^{c/c}$;BGLAP-Cre* mice, *BGLAP-Cre;Rosa26$^{mT/mG}$* mice and control littermates were anesthetized and the left-leg Tibialis anterior tendon was accessed through the skin. longitudinal incisions were made along the medial and lateral borders of the tendon. Jewelers forceps were then slid under the tendon and spread to tension the tendon. The lateral edge of the defect was created with a scalpel and then jewelers forceps were placed into the incision and pushed through the tendon to create a full-thickness medial edge of the defect. Incisions were closed with prolene suture (Ethicon, USA). For the sham procedure in the contralateral limb, longitudinal incisions along the border were made and forceps were placed under the tendon but no defect was made. The animals were sacrificed 1, 2 or 4 weeks after injury.

## Sheath transplantation

The left-leg Achilles tendons of 4 month-old female immunocompromised mice were injured as described above except that this tendon was not surrounded by true sheath tissue. Immediately after injury, 2 mg Tibialis anterior tendon sheath tissues (approximately two pieces of sheath tissues) were harvested from the *BGLAP-Cre;Rosa26$^{mT/mG}$* mice or the *Ptch1$^{LacZ/+}$* mice and placed underneath the injured Achilles tendon of the immunocompromised mice. The animals were sacrificed for *BGLAP-Cre* lineage tracing analysis at specific time points from 2 days to 45 days. The tendon tissues were harvested for FACS-sorting and for molecular, histological, and biomechanical analysis at day 14. Sham operation was performed without transplantation.

## *In vivo* differentiation

Sheath-derived cells were transplanted subcutaneously to the dorsal surface or calvaria of 16-week-old female immunocompromised mice. Approximately $1 \times 10^6$ cells were cultured until confluent to form a continuous sheet in a 10 cm culture dish. The cell sheets were lifted by tweezers and rolled up to form a transplant. The transplants were then deposited subcutaneously to the dorsal surface of the mice for 8 weeks. For calvaria transplantation, approximately $1 \times 10^6$ cells were cultured until confluent. After trypsinization, the cells were mixed with Matrix gel (BD Biosciences, USA) and subcutaneously injected into the calvaria. The animals were sacrificed and analyzed 8 weeks after transplantation.

## Collagen content analysis

Procedures were followed as described previously (*Guerquin et al., 2013*) using the Hydroxyproline Assay Kit (Sigma-Aldrich, USA, MAK008). Briefly, mouse Tibialis anterior tendons or Achilles tendons were collected and homogenized. A hydroxyproline assay kit was used to measure the production of hydroxyproline in xenografted or non-grafted tendons by the reaction of oxidized hydroxyproline with dimethylaminobenzaldehyde (DMAB), which resulted in a colorimetric product (560 nm) that was proportional to the hydroxyproline quantity. Four independent samples per group were collected and analyzed.

## Immunohistochemistry and immunocytochemistry

Tissue sections or primary cells were fixed and blocked with 5% Donkey serum (Sigma-Aldrich, USA). Samples were then incubated with primary antibodies at 4°C overnight. The following antibodies were used for staining: Bglap (Abcam, USA, ab93876, 5 ug/ml); Tppp3 (Abcam, USA, ab150998, 1:50); Mkx (Lifespan Biosciences Inc., USA, LS-B8063, 1 ug/ml); Gli1 (R&D, USA, MAB3324, 10 ug/ml); Gli2 (R&D, USA, AF3635, 5 ug/ml); pH3 (Santa Cruz, USA, sc-8656-R, 1:200); Donkey anti-rabbit IgG (H + L), Alexa Fluor 594 conjugate (Thermo Fisher Scientific, USA, A21207, 1:500); Donkey anti-rat IgG (H + L), Alexa Fluor 488 conjugate (Thermo Fisher Scientific, USA, A21208, 1:500); Donkey anti-goat IgG (H + L), Alexa Fluor 488 conjugate (Thermo Fisher Scientific, USA, A11055, 1:500); and Goat anti-rabbit IgG (H + L), Alexa Fluor 633 conjugate (Thermo Fisher Scientific, USA, A21072, 1:500). Samples were washed with PBS and incubated with secondary antibodies at room temperature for 1 hr. Nuclei were stained with Hoechst 33342 (Thermo Fisher Scientific, USA). Signals were detected under fluorescence microscopy. For TUNEL assay, procedures were followed as described by the manufacturer using the ApopTag Plus In Situ Apoptosis Fluorescein Detection Kit (Millipore, USA, s7111).

## Cell isolation and culture

Mouse Tibialis anterior sheaths and tendons from 8–10-week-old mice were dissected separately, minced and digested with 3 mg/ml collagenase type I (Worthington, USA) and 4 mg/ml dispase (Thermo Fisher Scientific, USA) in PBS for 1 hr at 37°C. Single cell suspension was then cultured in 20% FBS, 100 mM 2-mercaptoethanol and 1% penicillin-streptomycin (Thermo Fisher Scientific, USA) in α-MEM (Thermo Fisher Scientific, USA) for 8–10 days. All the experiments were performed with primary cells of passage one.

## FACS analysis

Primary cells isolated from the tendon and sheath tissues of the *BGLAP-Cre;Rosa26^{mT/mG}* mice and primary cells from the tendon tissues of the immunocompromised mice with *BGLAP-Cre;Rosa26^{mT/mG}* sheath transplantation were sorted and analyzed for GFP or tdTomato fluorescence using FACS Aria Fusion flow cytometer (BD Biosciences, USA). Data were calculated using the FACSDiva Software (Histogram) (BD Biosciences, USA).

## Colony formation assay and *in vitro* multipotent differentiation

For colony formation assay, single cell suspensions of sorted tendon sheath-derived cells (GFP$^+$) or tendon fiber-derived cells (tdTomato$^+$) from the *BGLAP-Cre;Rosa26^{mT/mG}* mice were cultured in a 60-mm dish for 14 days at two different cell densities (1000 or 2000). *In vitro* multipotent differentiation towards osteogenesis and adipogenesis was performed as previously described (*Mak et al., 2008a*; *Rutigliano et al., 2013*). Briefly, approximately $1 \times 10^5$ cells were seeded per well in 24-well plates in α-MEM supplemented with 20% FBS and 1% penicillin-streptomycin (Thermo Fisher Scientific, USA). 12~24 hr later, the medium was switched to osteogenic/adipogenic differentiation medium after reaching superconfluence. For osteogenesis, the differentiation cocktail includes 100 ug/ml ascorbic acid and 10 mM glycerophosphate (Sigma-Aldrich, USA). For adipogenesis, the differentiation cocktail contains 10 μg/ml insulin, 100 nM dexamethasone, 250 μM IBMX and 200 μM indomethacin (Sigma-Aldrich, USA). The medium was changed every 3 days. The cells were examined for osteogenic or adipogenic differentiation 12 days after the induction. Osteogenic differentiation was stained by Alizarin Red S staining. Adipogenic differentiation was stained by 0.3% Oil Red O staining (Sigma-Aldrich, USA). For chondrogenesis, a high-density micromass culture system was

applied as described previously with modification (*Zhang et al., 2010*; *Guzzo et al., 2013*; *Thakor et al., 2011*; *Neupane et al., 2008*). Briefly, cells were harvested and resuspended in chondrogenic medium at $2.5 \times 10^6$ cells/50 µl. Droplets (10 µl) were carefully placed in each well of a 24-well plate. Cells were allowed to adhere at 37°C for 2~4 hr, followed by the addition of 500 µl chondrogenic medium. The medium was changed every 3 days. Micromass cultures were stained or harvested for molecular analysis at day 21. Chondrogenic differentiation cocktail includes 1% insulin-transferrin-selenium solution (ITS, Thermo Fisher Scientific, USA), 10 ng/ml TGF-β1 (Peprotech, USA), 100 nM dexamethasone (Sigma-Aldrich, USA), 50 µg/ml ascorbic acid (Sigma-Aldrich, USA), and 1 mM sodium pyruvate (Thermo Fisher Scientific, USA). Chondrogenic differentiation was stained by Alcian Blue or Safranin O staining.

## Quantitative RT-PCR

RNA was reverse-transcribed to cDNA using the M-MLV Reverse Transcriptase (Thermo Fisher Scientific, USA). cDNA was analyzed by qPCR using Power SYBR Green PCR Master Mix (Applied Biosystems, USA). Relative mRNA levels were calculated using the comparative CT method and normalized to *beta-tubulin* or *Gapdh* mRNA. Primer sequences for QRT-PCR were shown in *Supplementary file 1*.

## Dual-luciferase reporter assay

The 0.8 kb and 4060 bp Mkx promoter sequence were amplified from mouse genomic DNA by PCR using the following primers: *Mkxp8K* forward primer, 5'- GGGGTACCACTGGAAATGGCTTTATTG TAT-3'; *Mkxp8K* reverse primer, 5'- CTAGCTAGCGCGGCTGACACTCCTGT-3'; *Mkx4K* forward primer, 5'- CTAGCTAGCGGATGGAGGCTGGAGAACTTG-3'; and Mkx4K reverse primer, 5'-CCCAAGCTTGCGGCTGACACTCCTGTC-3'. The wild-type *Mkx* promoter regions were cloned into a *pGL3*-basic reporter vector using the KpnI and NheI sites for the 855-bp fragment and the XhoI and HindIII sites for the 4060-bp fragment. HEK-293T cells were co-transfected with 100 ng pcDNA3.1(+)/myc-His A (Thermo Fisher Scientific, USA), pCMV5SMAD2-HA or pCMV5B-Flag-SMAD3, 100 ng reporter plasmids and 5 ng pRL-TK plasmid (Promega, USA) using Lipofectamine 2000 (Thermo Fisher Scientific, USA) according to the manufacturer's instructions. 48 hr after transfection, firefly and Renilla luciferase activities were assayed using the Dual-luciferase reporter assay system (Promega, USA) according to the manufacturer's protocol. pCMV5 SMAD2-HA was a gift from Joan Massague (Addgene plasmid # 14930) (*Hata et al., 1997*). pCMV5B-Flag-SMAD3 was a gift from Jeff Wrana (Addgene plasmid # 11742) (*Labbé et al., 1998*). The activity of the firefly luciferase was normalized to Renilla luciferase activity.

## Western blotting

Sheath tissues or cells were homogenized in cold RIPA lysis buffer supplemented with protease inhibitor cocktails (Roche, Switzerland) and phosSTOP phosphatase inhibitor (Roche, Switzerland). Protein concentration was determined by BCA assay (Thermo Fisher Scientific, USA). 20 µg protein samples were separated by SDS-PAGE and transferred to PVED membrane. After blocking with 5% BSA in $1 \times$ TBST buffer, membranes were hybridized with the respective primary antibodies: Mkx (Abcam, USA, ab179597, 1 ug/ml), Collagen I (Abcam, USA, ab292, 1:1000), TGF-b1 (Abcam, USA, ab64715, 1:500), Smad3 (CST, USA, #9523, 1:1000), p-Smad3 (CST, USA, #9520, 1:1000), or β-Actin (CST, USA, #8457, 1:10000) overnight. Then, the membranes were incubated with HRP-conjugated secondary anti-rabbit IgG (1 : 5000), anti-mouse IgG (1 : 5000) or anti-rat IgG (1 : 5000) for 2 hr at room temperature. Signals were visualized with the ECL system (PerkinElmer, Waltham, MA, USA).

## Biomechanical testing

Biomechanical testing was performed as previously described with modifications (*Chan et al., 1998*; *Ni et al., 2012*, *2013*). The Achilles tendon-foot composite was prepared and isolated from the hind limb. The composite was fixed on the testing jig with two clamps. The lower clamp was used to fix the Achilles tendon while the upper one was used to fix the distal foot. The Achilles tendon-foot composite was then loaded to a Mach-1 micromechanical system (Biomomentum Inc., Canada) for testing. The 'test to failure' was performed at a testing speed of 0.1 mm/s and preload of 0.5N using a 70 N load cell. The load-displacement curve of the healing tendon tissue was recorded. Peak stress

(N/mm$^2$) was calculated based on Peak load divided by the cross-sectional area (Peak stress (N/mm$^2$) = Peak force (N)/ [π *(diameter/2)$^2$] (mm$^2$)). Tendon stiffness was calculated by dividing load to failure by tendon deflection [N/mm] (*Palmes et al., 2002*).

## Statistical analysis

No statistical methods were used to predetermine sample size. The experiments were not randomized and the investigators were not blinded to allocation during experiments and outcome assessment. Prism software (GraphPad, USA) was used for all statistical analyses. Statistical comparisons were performed using an unpaired two-tailed Student's t-test. *P* values were considered significant when less than 0.05. One-way or two-way ANOVA followed by Tukey's test was used for multiple groups' comparison as discussed for specific experiments. Results represent means and error bars indicate s.e.m. Equal variances were assumed. At least three independent experiments were performed for all biochemical experiments and representative images were shown.

## Human specimens

Tendon grafts were collected from young adults (aged between 18–40, both sexes) with patellar tendinopathy or from control subjects undergoing anterior cruciate ligament reconstruction with no previous history or clinical signs of patellar tendon injury and tendinopathy. Specimen collection was approved by the Clinical Research Ethics Committee of the Chinese University of Hong Kong (2013.479). Informed consents were obtained from all participants.

## Acknowledgements

We thank Mr. Bruma Fu, Department of Orthopaedics and Traumatology, The Chinese University of Hong Kong for coordination of human specimen collection. This work is supported by the Seed Fund of the School of Biomedical Sciences, The Chinese University of Hong Kong (4620504) and a donation made by Lui Che Woo Foundation Limited to KKM.

## Additional information

### Funding

| Funder | Grant reference number | Author |
| --- | --- | --- |
| Chinese University of Hong Kong | 4620504 | Kingston King-Lun Mak |
| Lui Che Woo Foundation | | Kingston King-Lun Mak |

The funders had no role in study design, data collection and interpretation, or the decision to submit the work for publication.

### Author contributions

Yi Wang, Conceptualization, Data curation, Formal analysis, Supervision, Funding acquisition, Investigation, Methodology, Writing—original draft, Project administration, Writing—review and editing; Xu Zhang, Conceptualization, Data curation, Formal analysis, Validation, Visualization, Methodology, Writing—original draft, Project administration; Huihui Huang, Data curation, Formal analysis, Methodology; Yin Xia, Data curation, Formal analysis; YiFei Yao, Resources, Methodology; Arthur Fuk-Tat Mak, Resources, Data curation, Methodology; Patrick Shu-Hang Yung, Kai-Ming Chan, Li Wang, Kingston King-Lun Mak, Resources; Chenglin Zhang, Yu Huang, Resources, Data curation

### Author ORCIDs

Yi Wang http://orcid.org/0000-0002-0291-5314
Kingston King-Lun Mak http://orcid.org/0000-0002-4733-9146

## Ethics

Human subjects: Specimen collection was approved by the Clinical Research Ethics Committee of the Chinese University of Hong Kong (2013.479). Informed consents were obtained from all participants.

Animal experimentation: All experiments were approved by the Animal Experimentation Ethics Committee of the Chinese University of Hong Kong (16-637).

## Decision letter and Author response

Decision letter https://doi.org/10.7554/eLife.30474.052
Author response https://doi.org/10.7554/eLife.30474.053

## Additional files

### Supplementary files

• Supplementary file 1. Primer sequences for QRT-PCR.
DOI: https://doi.org/10.7554/eLife.30474.050

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
