## [Decision Letter]

Thank you for submitting your article "Osteocalcin expressing cells from tendon sheaths contribute to tendon repair by activating Hedgehog signaling" for consideration by *eLife*. Your article has been favorably evaluated by Didier Stainier (Senior Editor) and three reviewers, one of whom is a member of our Board of Reviewing Editors. The following individual involved in review of your submission has agreed to reveal his identity: Henry Kronenberg (Reviewer #2).

The reviewers have discussed the reviews with one another and the Reviewing Editor has drafted this decision to help you prepare a revised submission.

The reviewers agree that this is an interesting paper showing that tendon sheath cells express osteocalcin and have stem-like properties in vitro, including differentiation into cells with tendon-like gene expression. After transplantation, these cells clearly associate with the tendon, which correlates with upregulation of tendon-associated genes and improved mechanical properties of the injured tendon. While the reviewers note the many strengths of this substantial study, a major and unanimous concern was that the assertion that tendon sheath cells contribute to new tenocytes was not well enough substantiated. For example, the data in Figure 3 do not conclusively show that GFP^+^ sheath cells are expressing a molecular tendon signature, as opposed to inducing neighboring host cells to induce these markers. An alternative explanation that cannot be excluded is that sheath cells contribute instead to scarring around the tendon, as sheath-mediated scarring is frequently considered a major clinical problem after laceration injuries (see studies by the labs of Richard Gelberman, Hani Awad, and Regis O'Keefe). Additional experiments are therefore required to demonstrate that HOC-Cre/GFP^+^ sheath cells are differentiating into tenocytes after injury. This may involve FACS-sorting GFP^+^ cells after the graft in Figure 3 and performing molecular analysis and/or co-staining GFP^+^ cells in sections with tendon markers (e.g. Mkx). It would also be helpful to demonstrate that HOC-Cre/GFP^+^ cells contribute to new tenocytes after tendon injury (without grafting), for example by showing GFP^+^ cells within the tendon in cross-section (i.e. Figure 1) and/or co-staining anti-GFP with anti-Mkx or other tendon markers after tendon repair following injury.

The reviewers also identified a number of more minor issues that should be addressed to improve the manuscript - listed below. The manuscript would also benefit from careful proofreading as there are a number of grammatical and spelling errors. Given that these new experiments may take some time, it may be helpful to hear what your plan will be to address these concerns and your expected time-frame, especially related to the major concern about HOC-Cre lineage tracing mentioned above.

1. The Mohawk antibody appears to stain beyond the nucleus, which is not typical for transcription factors (Figure 6 and Figure 7). It is also curious that Mkx is not observed in tendon cells since this is typically considered a tendon differentiation marker. Please explain. Mohawk in situ hybridization data for at least some of these experiments could help confirm results.

2. For histology (Figure 1, Figure 3), native tendon controls should be included for comparison.

3. In their Discussion, the authors talk about how osteocalcin protein is likely to contribute to the tendon matrix proteins. While this is likely to be true, the authors should mention that osteocalcin is a secreted hormone that binds to receptors (one of which is called GPRC6A) and has been shown to have effects in mice on islets secreting insulin, and on fat, muscle, and nerve cells (see for example, Mera et al, Cell Metabolism 23:1078(16)). It would be useful (but not required here) to see if tendon cells, particularly after injury, express receptors for osteocalcin, and, of course, it would be of interest to know if osteocalcin KO mice have normal tendon repair. None of this needs to be shown for this paper, however. But the possible paracrine actions of osteocalcin should be mentioned.

4. The authors show that 80% of cells isolated from sheath tissues are HOC-Cre labeled and then use these cells to derive colonies. What percentage of these cells actually have clonogenic capacity? The authors also show considerable proliferation in sheath cells even at adult stages, yet no apoptosis - where are these cells going and how are they turned over?

5. The methods for their multipotency assays should be described in more detail since the papers referenced do not actual give sufficient detail about this (in the Mak paper only chondrogenic differentiation of micromass is described and this protocol would not work for adult stem cell sources such as MSCs). The authors show that TGFβ does not induce Sox9 or Col2 in Figure 2—figure supplement 1; TGFβ is the typical growth factor used to induce chondrogenesis in multipotency assays so it is unclear how this was done in Figure 2 and H.

6. Methods for transplantation of sheath tissue should be described in more detail - how much tissue was transplanted (how many pieces? what size pieces?), how was this kept consistent between experiments?

7. What were the stiffness or modulus values (Figure 3)? Were these also improved with transplantation?

8. Transient increase in control sheath thickness between 2-4 months seems strange, please explain (Figure 6). Representative histology in Figure 6 does not show this jump.

9. This point in the discussion on osteocalcin is not quite logical: "The tendon/ligament and bone are in close proximity with each other and they are integral parts of the skeletal system, it is not surprising that Osteocalcin is also found in tendon-related tissues." There are many tissues in close proximity that do not share markers.

10. The human tendinopathy results in Figure 9 seem to contradict the notion that activation of HH signaling and Mkx are indicators of a regenerative response since human tendons do not regenerate after tendinopathy/degeneration or acute injury. While they may be mounting a 'repair' response, it is not regeneration and perhaps shows that these are indicators of disease.

11. Figure 8 should be a separate figure. Panel F is too small. Also, its unclear how the -714 and -156bp regions correspond to the 4bk and 0.8kb regions discussed below. This was confusing.

12. The discussion of the inflammatory response is speculative and should be justified or removed. Hh activation could also be due to cell death, mechanical forces, etc.

13. In many instances, "duplicates" should be replaced by "replicates" (i.e. when n is greater than 2).

14. In Figure 2—figure supplement 1, panels D-G are not obviously tendon-like as in Figure 2.

15. Need to explain more clearly that conditional loss of Ptch1 leads to gain of LacZ.

16. Figure 5—figure supplement 1, would be helpful to show individual green/red channels in (C).

17. Its unclear whether Mkx protein is up in Ptch1-c/c in Figure 8.

18. Figure 9 - What do the different rows means? Is each from a separate patient? Do the "adjacent tendon" domains in B correspond to the same "tendinopathy" patients? Does n=7 refer to 7 healthy and 7 tendinopathy patients?

[Editors' note: further revisions were requested prior to acceptance, as described below.]

Thank you for resubmitting your work entitled "Osteocalcin expressing cells from tendon sheaths in mice contribute to tendon repair by activating Hedgehog signaling" for further consideration at *eLife*. Your revised article has been favorably evaluated by Didier Stainier (Senior Editor) and a Reviewing Editor.

The manuscript has been improved but there are some remaining issues that need to be addressed before acceptance, as outlined below:

1. The new data in Figure 3—figure supplement 1 and 2 now show convincingly that Bglap^+^ cells are located in the tendon proper following injury. While these cells express Mkx and Col1a1, the experiments fall short of showing that these are bona fide tenocytes. Mkx is also highly expressed in the tendon sheath in un-injured controls, and thus it is unclear whether Mkx is solely a marker for tendon progenitor cells, or alternatively also for non-tenocyte sheath cells. Likewise, Col1a1 is a widely expressed collagen gene and not a very specific marker. It therefore seems premature to conclude that the Mkx^+^, Col1a1^+^ cells within the tendon after injury are bona fide tenocytes, as they may alternatively be scarring sheath cells that migrate into the tendon proper. While new experiments are not required, this caveat should be well described in the Discussion – i.e. acknowledging that it remains to be determined the extent to which these sheath-derived Bglap^+^ lineage cells are bona fide tenocytes, as opposed to scarring sheath cells that have invaded into the injured tendon.

2) The conclusions of the human data in Figure 10 should be further toned down, acknowledging that it remains unclear whether Mkx/Gli1 expression represents a scarring and/or tendon repair process.

3) The new data in Figure 3—figure supplement 1 and 2 are critical to the paper and should be included in the main Figure 3. It would be sufficient to move only the merged images to main Figure 3, with the individual channels remaining as supplements. The expression data in Figure 3—figure supplement 2 should also be moved to main Figure 3. It would also be helpful to show a merged image of just the GFP and Mkx channels (Figure 3—figure supplement 1) as the merge of all channels is not easy to appreciate the overlap of GFP and Mkx.

4) In Figure 3—figure supplement 2, "Relative expression level" is relative to what? A source data file should be included for this supplement, and what is meant by the "relative expression level" should be better described in the figure legend.

5) "It has been previously shown that Mkx promotes tenogenesis through activation of TGFβ signaling (Liu et al., 2015), thus we searched for Smad binding sites in the promoter regions of Mkx." This sentence is confusing as first part implies Mkx upstream of Tgfβ, but then experiment addresses Tgfβ/Smad3 upstream of Mkx.

6) "Conditional loss of Ptch1 leads to gain of LacZ expression." Need to further clarify what "conditional loss" means in this context. In the germline – i.e. embryowide?

7) "We established that Bglap-expressing cells from sheath tissues possess stem/progenitor cell properties and they are required for tendon repair." The data do not formally show a requirement for Bglap^+^ cells in tendon repair, as this would require their ablation. It may be better to say that the data show that they participate in tendon repair.

---

## [Author Response]

The reviewers agree that this is an interesting paper showing that tendon sheath cells express osteocalcin and have stem-like properties in vitro, including differentiation into cells with tendon-like gene expression. After transplantation, these cells clearly associate with the tendon, which correlates with upregulation of tendon-associated genes and improved mechanical properties of the injured tendon. While the reviewers note the many strengths of this substantial study, a major and unanimous concern was that the assertion that tendon sheath cells contribute to new tenocytes was not well enough substantiated. For example, the data in Figure 3 do not conclusively show that GFP^+^sheath cells are expressing a molecular tendon signature, as opposed to inducing neighboring host cells to induce these markers. An alternative explanation that cannot be excluded is that sheath cells contribute instead to scarring around the tendon, as sheath-mediated scarring is frequently considered a major clinical problem after laceration injuries (see studies by the labs of Richard Gelberman, Hani Awad, and Regis O'Keefe). Additional experiments are therefore required to demonstrate that HOC-Cre/GFP^+^sheath cells are differentiating into tenocytes after injury. This may involve FACS-sorting GFP^+^cells after the graft in Figure 3 and performing molecular analysis and/or co-staining GFP^+^cells in sections with tendon markers (e.g. Mkx).

We thank the reviewers for these comments and suggestions. The expression of tendon progenitor marker Mkx was examined in the injured tendon sections (with/without sheath transplantation groups) as shown in newly added Figure 3—figure supplement 2. The Mkx expression was detected with the secondary antibody with far-red fluorescence (Excitation Maximum 633nm, A-21072, Thermo Fisher Scientific, USA) to avoid the confusion with tdTomato signals (Excitation Maximum, 554nm). Our results showed that some of the GFP^+^ cells expressed Mkx in the areas of the injured Achilles tendon tissues. Furthermore, FACS-sorted GFP^+^ cells isolated from the graft in Figure 3 by qRT-PCR as shown in newly added Figure 3—figure supplement 2 also showed upregulation of tendon progenitor marker *Mkx* and main ECM component *Col1a1* as compared to the control sheaths. These results indicate that sheath cells at least in part differentiate into tenocytes after injury, contributing to tendon repair. We do agree that sheath cells may also contribute to scarring around the injured tendons, which cannot be excluded from our data.

It would also be helpful to demonstrate that HOC-Cre/GFP^+^cells contribute to new tenocytes after tendon injury (without grafting), for example by showing GFP^+^cells within the tendon in cross-section (i.e. Figure 1) and/or co-staining anti-GFP with anti-Mkx or other tendon markers after tendon repair following injury.

The distribution of the GFP^+^ cells has been examined within tendon tissues during tendon repair following injury in *BGLAP-Cre;Rosa26^mT/mG^*mice (without grafting). Both cross-section and longitudinal-section were provided in the newly added Figure 3—figure supplement 1. Noticeably, we found considerable GFP^+^ cells appeared in the injured tendon fibers as shown from both cross- and longitudinal sections. Furthermore, the GFP^+^ cells co-expressed tendon progenitor marker Mkx according to the immunofluorescent staining as shown in the newly added Figure 3—figure supplement 1. These data suggest that sheath tissues differentiate into Mkx-expressing progenitor cells and contribute to tendon repair.

[…] 1. The Mohawk antibody appears to stain beyond the nucleus, which is not typical for transcription factors (Figure 6 and Figure 7). It is also curious that Mkx is not observed in tendon cells since this is typically considered a tendon differentiation marker. Please explain. Mohawk in situ hybridization data for at least some of these experiments could help confirm results.

To improve the specificity of Mkx expression, we purchased new Mkx antibodies from two companies (LS B8063, Lifespan Biosciences Inc, USA; ab179597, Abcam, USA). We optimized the condition with new dilution of antibodies (LS B8063, 1 ug/ml; ab179597, 1μg/ml) as mentioned in the revised Materials and methods. The specificity of Mkx antibody was greatly improved and the Mkx signals mainly appeared in the nucleus as shown in the revised Figure 5, Figure 6, Figure 7. Our data indicate that the expression of tendon progenitor marker Mkx was positively correlated with the activation of Hh signaling in sheath cells in injured condition as shown by our gain-of-function and loss-of-function knockout mouse models.

As a progenitor marker, the expression of Mkx is relatively low in postnatal tendon tissues as previously described (Liu, Watson et al. 2010). Strong Mkx expression is observed in the tendon progenitors at E13.5 and E14.5 (Anderson, Arredondo et al. 2006, Liu, Watson et al. 2010) and subsequently declined in differentiated tendon cells in the limb and tail tendons at E16.5. The expression of Mkx is very weak in tendon cells at postnatal stages P0 and P14. These findings are consistent with our observations that the expression level of Mkx in mature tendon tissues is very low as compared to the newly differentiated tendon progenitor cells.

2. For histology (Figure 1, Figure 3), native tendon controls should be included for comparison.

We included the native tendon controls in the revised Figure 1 and Figure 3 as suggested.

3. In their Discussion, the authors talk about how osteocalcin protein is likely to contribute to the tendon matrix proteins. While this is likely to be true, the authors should mention that osteocalcin is a secreted hormone that binds to receptors (one of which is called GPRC6A) and has been shown to have effects in mice on islets secreting insulin, and on fat, muscle, and nerve cells (see for example, Mera et al, Cell Metabolism 23:1078(16)). It would be useful (but not required here) to see if tendon cells, particularly after injury, express receptors for osteocalcin, and, of course, it would be of interest to know if osteocalcin KO mice have normal tendon repair. None of this needs to be shown for this paper, however. But the possible paracrine actions of osteocalcin should be mentioned.

We thank for the comments. We have included the paracrine actions of Osteocalcin in the first paragraph of the Discussion.

4. The authors show that 80% of cells isolated from sheath tissues are HOC-Cre labeled and then use these cells to derive colonies. What percentage of these cells actually have clonogenic capacity? The authors also show considerable proliferation in sheath cells even at adult stages, yet no apoptosis - where are these cells going and how are they turned over?

We found that approximately 1.5% of BGLAP-Cre labeled cells have clonogenic capacity according to colony formation assay as shown in the revised Figure 1—figure supplement 1. We repeated the staining of sheath sections of the *Ptch1^c/c^; BGLAP-Cre* and their 4 littermates with a new ApopTag Plus Fluorescein In Situ Kit (S7111) (Millipore, USA). There were only very few apoptotic cells in the sheaths of the *Ptch1^c/c^; BGLAP-Cre* mice as shown in the revised Figure 5. Both high proliferation and low apoptotic rates contribute the increase sheath thickness observed in the *Ptch1^c/c^; BGLAP-Cre* mice. The turnover of these cells is expected to be relatively long as compared to other tissues.

5. The methods for their multipotency assays should be described in more detail since the papers referenced do not actual give sufficient detail about this (in the Mak paper only chondrogenic differentiation of micromass is described and this protocol would not work for adult stem cell sources such as MSCs). The authors show that TGFβ does not induce Sox9 or Col2 in Figure 2—figure supplement 1; TGFβ is the typical growth factor used to induce chondrogenesis in multipotency assays so it is unclear how this was done in Figure 2 and H.

The details of multipotent differentiation are as follows and we have added a more detailed description to the revised Materials and methods section “Colony formation assay and in vitro multipotent differentiation”:

“In vitro multipotent differentiation towards osteogenesis and adipogenesis was performed as described (Mak, Bi et al. 2008, Rutigliano, Corradetti et al. 2013). […] Chondrogenic differentiation cocktail includes 1% insulin-transferrin-selenium solution (ITS, Thermo Fisher Scientific, USA), 10ng/ml TGF-β1 (Peprotech, USA), 100nM dexamethasone (Sigma-Aldrich, USA), 50μg/ml ascorbic acid (Sigma-Aldrich, USA), 1 mM sodium pyruvate (Thermo Fisher Scientific, USA).”

In our study, higher cell density at 2.5x106 cells/50μl was used. Our results showed that this micromass culture system also works in chondrogenesis using sheath-derived stem/progenitor cells.

For BMP2 and TGF-β1 treatment, primary sheath cells were serum starved with 2% FBS overnight, then treated with 100 ng/ml BMP2 (CST, USA) or 2ng/ml TGF-β1 (PeproTech, USA). Cells were collected after 48 h treatment. We agree that TGF-β1 should induce Sox9 or Col2a1 expression as a typical growth factor used for chondrogenesis. We repeated the experiment with the treatment of new aliquots of the recombinant proteins (BMP2 and TGF-β1) and re-examined the gene expression profiles of Sox9 and Col2a1. TGF-β1 indeed induced the expression of Sox9 and Col2a1 after such refinement as shown in the revised Figure 2—figure supplement 1.

6. Methods for transplantation of sheath tissue should be described in more detail - how much tissue was transplanted (how many pieces? what size pieces?), how was this kept consistent between experiments?

A more detail description has been added to the revised Materials and methods. Sheath tissue with same weight was transplanted for consistency among animals. We transplanted 2mg sheath tissue (approximately two pieces of sheath tissue) per injured tendon for the examination of tendon repair process. We have included this information to the Sheath Transplantation section of the revised Materials and methods.

7. What were the stiffness or modulus values (Figure 3)? Were these also improved with transplantation?

We analyzed the stiffness in the sheath transplantation experiment and added the data in the revised Figure 3. Briefly, the stiffness was decreased in the injured group as compared to the sham group. Importantly, this reduction was significantly rescued in the injured + sheath transplantation group. This data indicate that tendon repair is improved with sheath transplantation, which is consistent with our previous conclusion.

8. Transient increase in control sheath thickness between 2-4 months seems strange, please explain (Figure 6). Representative histology in Figure 6 does not show this jump.

We re-examined the thickness of sheath tissues in all mutant mice and their littermates with new samples. We provided the new sheath thickness data in the revised Figure 6. The results were now consistent for Figure 6 and B.

9. This point in the discussion on osteocalcin is not quite logical: "The tendon/ligament and bone are in close proximity with each other and they are integral parts of the skeletal system, it is not surprising that Osteocalcin is also found in tendon-related tissues." There are many tissues in close proximity that do not share markers.

We thank the reviewers for the suggestion. We agreed to remove this statement from the revised manuscript.

10. The human tendinopathy results in Figure 9 seem to contradict the notion that activation of HH signaling and Mkx are indicators of a regenerative response since human tendons do not regenerate after tendinopathy/degeneration or acute injury. While they may be mounting a 'repair' response, it is not regeneration and perhaps shows that these are indicators of disease.

We thank the reviewers for the comments. In this manuscript, all tendinopathy samples were collected from patients after surgery with acute injuries or ruptures. The expression of Mkx is significantly increased during tendon repair according to previous studies (Juneja 2013) and our data in this manuscript. We found that Hh signaling is activated after tendon injury and activation of Hh signaling upregulates the expression of Mkx. Co-localization of GLI1^+^MKX^+^

cells was observed in the human acute tendon injury specimen. Therefore,

activation of Hh signaling and induction of Mkx expression are closely associated with a 'repair' response after acute injury. We agree that the co-localization of GLI1^+^MKX^+^ cells is not an indicator of a regenerative response but indicates a 'repair' response in the human specimen after acute injury.

11. Figure 8 should be a separate figure. Panel F is too small. Also, its unclear how the -714 and -156bp regions correspond to the 4bk and 0.8kb regions discussed below. This was confusing.

We have re-arranged Figure 8 as a separate figure and adjusted the size of Panel F in the revised Figure 9 accordingly. There are three putative Smad binding element (SBE) motifs in the Mkx promoter: -4060, -714 and -156bp sites. For better presentation, we redrew the diagram corresponding to the three promoter regions of Mkx gene in the revised Figure 9.

12. The discussion of the inflammatory response is speculative and should be justified or removed. Hh activation could also be due to cell death, mechanical forces, etc.

We thank the reviewers for their comments. We have removed this part in the revised manuscript.

13. In many instances, "duplicates" should be replaced by "replicates" (i.e. when n is greater than 2).

We are sorry for the mistakes. We have replaced all misused "duplicates" with "replicates" in the figure legends and source data respectively.

14. In Figure 2—figure supplement 1, panels D-G are not obviously tendon-like as in Figure 2.

We agree with the reviewers that in Figure 2—figure supplement 1, panels D-G are not obviously tendon-like as in Figure 2. However, these tissues have indeed differentiated to some collagen-rich tissues according to (F) polarized light view and (G) Masson’s trichrome staining. Therefore, we replaced “tendon-like tissues” to “collagen-rich tissues” for a more appropriate description in the revised figure legend of Figure 2—figure supplement 1.

15. Need to explain more clearly that conditional loss of Ptch1 leads to gain of LacZ.

We validated the Hh signaling activity in vivo by using the *Ptch1^lacZ/+^* mouse models as described (Mak, Chen et al. 2006). A knockin sequence with *LacZ* reporter replaces the *Ptch1* gene sequence at the translation initiation site. Since *Ptch1* is a transcriptional target gene of Hh signaling, the expression of the *LacZ* reporter represents the activity of Hh signaling. This information is added to the Results section “Hh signaling promotes proliferation of sheath progenitor cells”.

16. Figure 5—figure supplement 1, would be helpful to show individual green/red channels in (C).

We added individual green/red channels to the revised Figure 5—figure supplement 1, 1D and 1E as suggested.

17. Its unclear whether Mkx protein is up in Ptch1-c/c in Figure 8.

We have repeated the experiment with a new Mkx antibody (ab179597, 1 μg/ml, Abcam, USA). The expression of Mkx was more obviously increased in the revised Figure 9.

18. Figure 9 - What do the different rows means? Is each from a separate patient? Do the "adjacent tendon" domains in B correspond to the same "tendinopathy" patients? Does n=7 refer to 7 healthy and 7 tendinopathy patients?

Each row is a separate subject. The two "adjacent tendon" domains in B were representative images of tendon tissues of individual "tendinopathy" patients. n=7 refers to 7 healthy subjects and 7 tendinopathy patients. We have revised this information in the figure legend of the revised Figure 10.

[Editors' note: further revisions were requested prior to acceptance, as described below.]

The manuscript has been improved but there are some remaining issues that need to be addressed before acceptance, as outlined below:1. The new data in Figure 3—figure supplement 1 and 2 now show convincingly that Bglap^+^ cells are located in the tendon proper following injury. While these cells express Mkx and Col1a1, the experiments fall short of showing that these are bona fide tenocytes. Mkx is also highly expressed in the tendon sheath in un-injured controls, and thus it is unclear whether Mkx is solely a marker for tendon progenitor cells, or alternatively also for non-tenocyte sheath cells. Likewise, Col1a1 is a widely expressed collagen gene and not a very specific marker. It therefore seems premature to conclude that the Mkx^+^, Col1a1^+^ cells within the tendon after injury are bona fide tenocytes, as they may alternatively be scarring sheath cells that migrate into the tendon proper. While new experiments are not required, this caveat should be well described in the Discussion – i.e. acknowledging that it remains to be determined the extent to which these sheath-derived Bglap^+^ lineage cells are bona fide tenocytes, as opposed to scarring sheath cells that have invaded into the injured tendon.

We thank the reviewers for the comments. We have acknowledged such a possibility in our revised Discussion (second paragraph).

2) The conclusions of the human data in Figure 10 should be further toned down, acknowledging that it remains unclear whether Mkx/Gli1 expression represents a scarring and/or tendon repair process.

We have toned down our conclusion in the Results section as suggested (subsection “Hh signaling induces Mkx and Collagen I expression through TGFβ/Smad3 signaling”, last paragraph).

3) The new data in Figure 3—figure supplement 1 and 2 are critical to the paper and should be included in the main Figure 3. It would be sufficient to move only the merged images to main Figure 3, with the individual channels remaining as supplements. The expression data in Figure 3—figure supplement 2 should also be moved to main Figure 3. It would also be helpful to show a merged image of just the GFP and Mkx channels (Figure 3—figure supplement 1) as the merge of all channels is not easy to appreciate the overlap of GFP and Mkx.

The new data of Figure 3—figure supplement 1 and 2 have been rearranged to main Figure 3 as suggested. The expression data in Figure 3—figure supplement 2 is also moved to main Figure 3. We also added the merged images of GFP/Mkx channels in Figure 3—figure supplement 1.

4) In Figure 3—figure supplement 2, "Relative expression level" is relative to what? A source data file should be included for this supplement, and what is meant by the "relative expression level" should be better described in the figure legend.

In Figure 3—figure supplement 2, "Relative expression level" is relative to the expression level of *β-tubulin.* The source data file has been added for the supplement 2B in the revised version (Figure 3—source data 5). Internal controls have been added to all related figure legends.

5) "It has been previously shown that Mkx promotes tenogenesis through activation of TGFβ signaling (Liu et al., 2015), thus we searched for Smad binding sites in the promoter regions of Mkx." This sentence is confusing as first part implies Mkx upstream of Tgfβ, but then experiment addresses Tgfβ/Smad3 upstream of Mkx.

This confusing statement is deleted in the revised manuscript.

6) "Conditional loss of Ptch1 leads to gain of LacZ expression." Need to further clarify what "conditional loss" means in this context. In the germline – i.e. embryowide?

We have added more details to clarify the conditional loss of *Ptch1.* Mouse with floxed *Ptch1* is crossed with the *Stra8-Cre* mouse line that will result in germline loss of *Ptch1* expression and gain of *LacZ* expression. The revised content can be found in the first paragraph of the subsection “Hh signaling promotes proliferation of sheath progenitor cells”.

7) "We established that Bglap-expressing cells from sheath tissues possess stem/progenitor cell properties and they are required for tendon repair." The data do not formally show a requirement for Bglap^+^ cells in tendon repair, as this would require their ablation. It may be better to say that the data show that they participate in tendon repair.

We have revised the statement as suggested in the last paragraph of the Introduction.